# Targeting of apoptosis gene loci by reprogramming factors leads to selective eradication of leukemia cells

Yajie Wang[1,2,3,4,9], Ting Lu[5,9], Guohuan Sun[1,2,3,9], Yawei Zheng[1,2,3,9], Shangda Yang[1,3], Hongyan Zhang[1,3], Sha Hao[1,2,3,6,7], Yanfeng Liu[1,3], Shihui Ma[1,2,3], Houyu Zhang[5], Yongxin Ru[1,3], Shaorong Gao[8], Kuangyu Yen[5]*, Hui Cheng[1,2,3,6,7]* & Tao Cheng [1,2,3,6,7]*

Applying somatic cell reprogramming strategies in cancer cell biology is a powerful approach to analyze mechanisms of malignancy and develop new therapeutics. Here, we test whether leukemia cells can be reprogrammed in vivo using the canonical reprogramming transcription factors-Oct4, Sox2, Klf4, and c-Myc (termed as OSKM). Unexpectedly, we discover that OSKM can eradicate leukemia cells and dramatically improve survival of leukemia-bearing mice. By contrast, OSKM minimally impact normal hematopoietic cells. Using ATAC-seq, we find OSKM induce chromatin accessibility near genes encoding apoptotic regulators in leukemia cells. Moreover, this selective effect also involves downregulation of H3K9me3 as an early event. Dissection of the functional effects of OSKM shows that Klf4 and Sox2 play dominant roles compared to c-Myc and Oct4 in elimination of leukemia cells. These results reveal an intriguing paradigm by which OSKM-initiated reprogramming induction can be leveraged and diverged to develop novel anti-cancer strategies.

[1] State Key Laboratory of Experimental Hematology, Beijing, China. [2] National Clinical Research Center for Blood Diseases, Tianjin, China. [3] Institute of Hematology and Blood Disease Hospital, Chinese Academy of Medical Sciences and Peking Union Medical College, Tianjin, China. [4] Department of Hematology, the First People's Hospital of Yunnan Province, Yunnan, China. [5] Department of Developmental Biology, School of Basic Medical Sciences, Southern Medical University, Guangzhou, China. [6] Center for Stem Cell Medicine, Chinese Academy of Medical Sciences, Tianjin, China. [7] Department of Stem Cell & Regenerative Medicine, Peking Union Medical College, Tianjin, China. [8] School of Life Sciences and Technology, Tongji University, Shanghai, China. [9] These authors contributed equally: Yajie Wang, Ting Lu, Guohuan Sun, Yawei Zheng. *email: kuangyuyen@smu.edu.cn; chenghui@ihcams.ac.cn; chengtao@ihcams.ac.cn

Transcription factor induced reprogramming holds enormous promise in regenerative medicine and offers a powerful tool for studying pathogenesis. Upon co-expression of four transcription factors (Oct4, Sox2, Klf4, and c-Myc, collectively termed "OSKM" factors), somatic cells can be reprogrammed into induced pluripotent stem cells (iPSCs). This de-differentiation process involves dramatic changes in both gene expression and the chromatin landscape[1,2]. Often, reprogramming occurs in two different phases, with somatic cells progressing to a partially reprogrammed intermediate state before fully committing to reprogramming[3]. During this process, somatic-specific genes are turned off first, accompanied by a general closing of the chromatin, followed by upregulation of pluripotency genes and chromatin opening[4]. Given the chromatin obstacle and the complexity of the reprogramming process, de-differentiation is an inherently slow (~14 days) and inefficient procedure that only 0.1–3% of initiating cells can become iPSCs[5,6].

The process of iPSC derivation shares many characteristics with cancer development. Similar to the two phases in reprogramming, oncogenesis also proceeds step-wise from normal to pre-malignant before transitioning to malignancy. Cells undergoing either reprogramming or oncogenesis need to overcome epigenetic barriers before acquiring new identities (iPSCs or cancer cells, respectively)[7]. During reprogramming, somatic differentiated cells acquire the properties of self-renewal along with unlimited proliferation and exhibit global alterations to the transcriptional and epigenetic programs, which are also critical events during carcinogenesis. Moreover, the metabolic switch to glycolysis that occurs during somatic cell reprogramming is similarly observed in tumorigenesis[8]. In fact, many reprogramming factors, including the OSKM factors, are known oncogenic or cancer-promoting factors[9]. A number of studies using reprogrammable mice reported the occurrence of teratomas in multiple organs after short-term OSKM induction[10,11]. Given these similarities, it has been suggested that reprogramming processes and cancer development may involve overlapping or at least partially overlapping mechanisms.

Generation of iPSCs from cancer cells preserves oncogenic mutations and serves as an instructive tool to study cancer development[12]. Leukemia cells have been reprogrammed in vitro into leukemia-iPSCs[13]. Although successful, this in vitro reprogramming efficiency is extremely low (less than 0.001%) and only works on embryonic stem cell culture system. Genomic instability and altered epigenetic modifications harbored in cancer cells were suggested to be the cause of this inefficiency[14]. Moreover, several groups reported that cancer-derived iPSCs exhibit reduced malignant features compared with their parental cancer cells[15–17]. Meanwhile, for normal cells, in vivo-derived iPSCs confer totipotency features that are absent from in vitro generated iPSCs and have the potential to further develop into tumors[11,18]. Here we address the important open question of what happens to the established or already transformed cancer cells (not the normal or pre-malignant cells) when treated with OSKM in vivo.

To achieve cancer-iPSC reprogramming in vivo, we take advantage of an established tetO promoter-controlled mouse system[13]. We feed leukemia mice with Dox-containing water to induce OSKM expression in vivo. Strikingly, transient in vivo OSKM induction eliminates leukemia cells and extends the lifespan of leukemia-bearing mice. Within our tested time window, transient in vivo OSKM induction activates apoptosis in MLL-AF9 acute myeloid leukemia (AML) cells when grown in their natural habitat, either in vivo or in in vitro hematopoietic medium culture, but not in normal hematopoietic stem/progenitor cells. During in vivo OSKM induction, we observe increased chromatin accessibility at genes encoding apoptotic regulators,

decreased H3K9me3 levels, and upregulation of histone demethylase Kdm3a. This OSKM-induced apoptosis phenotype can be mimicked when MLL-AF9 AML cells are treated with chaetocin, a small-molecule inhibitor of H3K9 methylation. Furthermore, OSKM sensitivity of leukemia cells can be partially rescued by inhibition of histone demethylase Kdm3a. Among the four reprogramming factors, Sox2 and Klf4 are the key mediators of OSKM-mediated leukemia cell eradication. Our findings suggest that this effective reprogramming-apart cell elimination phenomenon could be leveraged to develop novel cancer therapeutics.

## Results

**OSKM factors strikingly reduce leukemia cells in vivo.** To reprogram leukemia cells in vivo, we took advantage of our MLL-AF9-OSKM leukemia cells[13], called MLL-AF9-OSKM hereafter. The cells contain the human MLL-AF9 fusion gene inserted endogenously and the Yamanaka reprogramming factors (Klf4, c-Myc, Oct4, and Sox2) under control of the tetO promoter. MLL-AF9-OSKM cells were intravenously injected into the tail vein of sub-lethally irradiated recipient C57BL/6J mice. When MLL-AF9-OSKM cells grew to reach 10–15, 40–60, or 90% of bone marrow (BM), we started to feed the recipient mice with 1 mg/mL Dox-containing water for 7 days as previously described[10] (Fig. 1a). qRT-PCR analyses confirmed that OSKM expression in MLL-AF9-OSKM cells was gradually upregulated after Dox treatment (Supplementary Fig. 1a). Without Dox treatment, all recipient mice died within 20 days. Unexpectedly, continuous induction of OSKM for 7 days in recipient mice resulted in a high survival rate for at least 1 year following Dox withdrawal. The survival rate was inversely correlated to the initial percentage of MLL-AF9-OSKM cells in BM (Fig. 1b). Interestingly, although teratoma had been reported in a previous study[10], we did not observe the formation of teratoma in this study, perhaps due to differences in the experimental systems and protocols applied in both studies. Moreover, we applied the same strategy to two additional mouse models of leukemia, mice that received MLL-NRIP3 AML cells or Notch1 T-cell acute lymphoblastic leukemia (T-ALL) cells, and observed a similar high survival rate (Supplementary Fig. 1b).

As the MLL-AF9 fusion protein contains a GFP tag, we could follow the dynamic change of MLL-AF9-OSKM cells using flow cytometry (Fig. 1c). Within 4 days of Dox treatment, the amount of MLL-AF9-OSKM cells in the spleen dropped from 81% to a nearly undetectable level, whereas in the BM, it dropped from 84.5% to 13%. After 7 days of Dox treatment, almost all MLL-AF9-OSKM cells in both spleen and BM were eliminated (Fig. 1c). We further cross-evaluated the amount of live MLL-AF9-OSKM cells during OSKM induction using in vivo two-photon imaging (Fig. 1d). Within 7 days of Dox-treatment, the GFP[+] leukemia cells dropped dramatically, in agreement with the flow cytometry data. These results suggested that most leukemia cells did not progress toward the iPSC state and instead were unable to survive reprogramming.

Given that we did not observe any occurrence of leukemia relapse or other tumors within 1 year after Dox withdrawal, we anticipated that OSKM induction might reduce levels of leukemia stem cells (LSCs), a key cellular element in leukemia initiation, maintenance, relapse, and drug resistance[19,20]. To test this, we first performed a colony-forming cell (CFC) assay to assess leukemogenic cells in vitro. The leukemia cells from BM and spleens of Dox-treated mice showed defective colony formation, with fewer and smaller colonies, particularly at 3 days post-Dox induction (Supplementary Fig. 1c). Then we quantified LSCs (IL7Rα⁻Lin⁻cKit⁺Sca1⁻ [21]) in the leukemia-bearing mice following OSKM induction. LSC frequency decreased rapidly

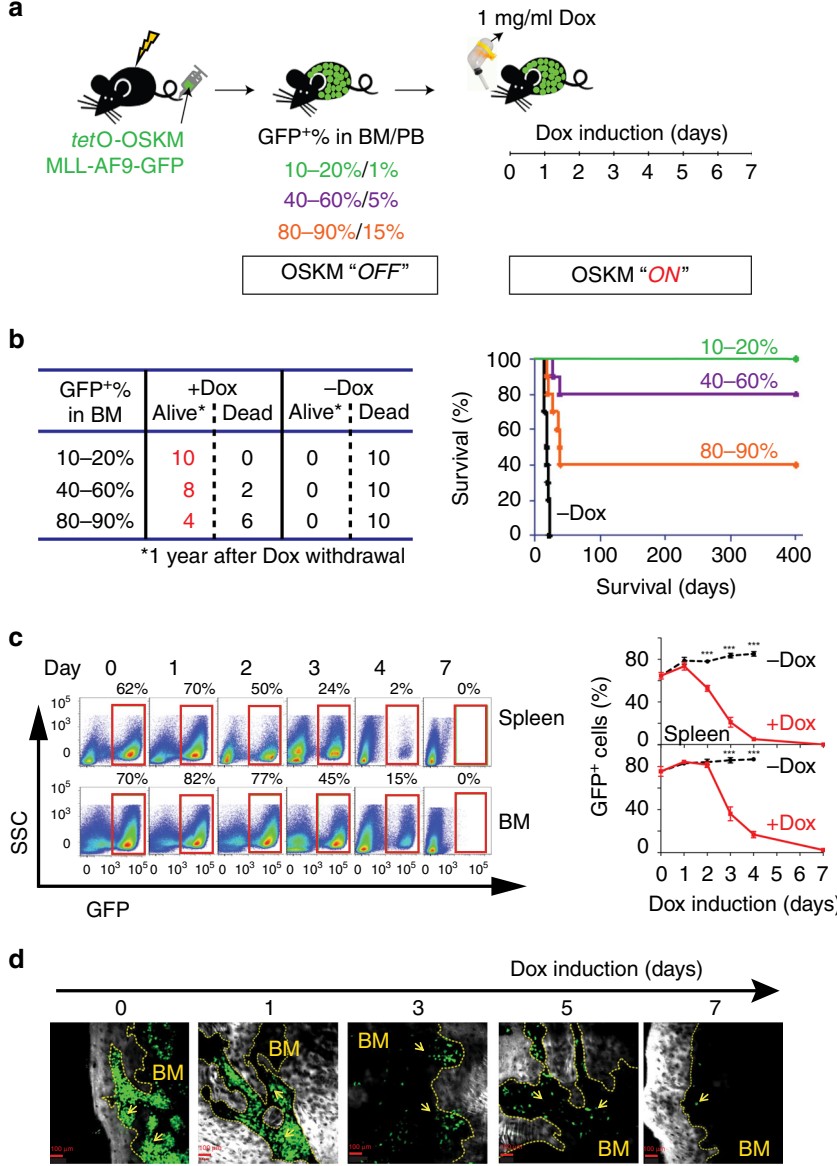

**Fig. 1 OSKM decreases leukemia cell burden. a** Schematic of the experimental procedure. Briefly, MLL-AF9-OSKM cells were injected into sub-lethally irradiated (4.5 Gy) recipient mice. When MLL-AF9-OSKM reached the indicated percentage in the bone marrow of the recipient mice, 1 mg/ml of doxycycline was added to drinking water for 7 days to induce OSKM expression. BM: bone marrow; PB: peripheral blood; Dox: doxycycline. **b** Left: One-year survival rate of recipient mice after Dox withdrawal; Right: Survival curves of recipient mice after Dox withdrawal. Black trace is the no Dox treatment control. Green, purple, and orange traces are the survival curves for recipient mice whose MLL-AF9-OSKM cells reached 10–20, 40–60, and 80–90% of bone marrow, respectively, before Dox treatment ($n = 10$). **c** Left: FACS analysis of spleen (upper) and bone marrow cells (lower) from recipient mice that underwent Dox treatment; Right: In vivo kinetics of MLL-AF9-OSKM cells (GFP$^+$) in spleen and bone marrow ($n = 3$–5, three independent experiments). ***$p < 0.001$, two-tailed Student's $t$ test. Error bars show SEM. **d** Representative two-photon images of leukemia cells (GFP$^+$) in the dorsal skull surface of leukemia-bearing mice. Scale bars, 100 μm. The yellow arrows indicate the GFP$^+$ leukemia cells.

after 1 day of Dox induction in both BM and spleen (Supplementary Fig. 1d). Moreover, as an independent functional assessment, a limiting dilution assay showed that the LSC frequency in the leukemia cell population was significantly decreased after OSKM induction (1/28,000 vs. 1/620, $p = 0.001$, Supplementary Fig. 1e). Together, these results suggest that ectopic expression of OSKM factors causes a strong reduction in leukemia cell number, especially LSCs, and that the remaining cells exhibit decreased proliferation and differentiation potential.

**OSKM factors have mild impact on normal hematopoietic cells.** To examine if this reduction is specific to leukemia cells, we used a competitive transplantation assay that compares the

proliferative capacities of two different populations of cells. We co-transplanted MLL-AF9-OSKM cells and BM cells from tetO-OSKM CD45.2$^+$ transgenic mice[22] into lethally irradiated recipient CD45.1$^+$ mice (Fig. 2a). When the amount of MLL-AF9-OSKM cells grew to reach 20% and 50% in BM of recipient CD45.1$^+$ mice, we started Dox treatment as described in Fig. 1a for 7 days. We first confirmed the ectopic expression of OSKM factors and noted that OSKM expression in MLL-AF9-OSKM cells was higher than that in tetO-OSKM Lin$^-$cKit$^+$Sca1$^+$ (LKS$^+$) hematopoietic stem/progenitor cells (HSPCs) but lower than that in Lin$^-$cKit$^+$Sca1$^-$(LKS$^-$) hematopoietic progenitor cells (HPCs) (Supplementary Fig. 2a). With Dox treatment, over 80% of the recipient mice were still viable 80 days after Dox

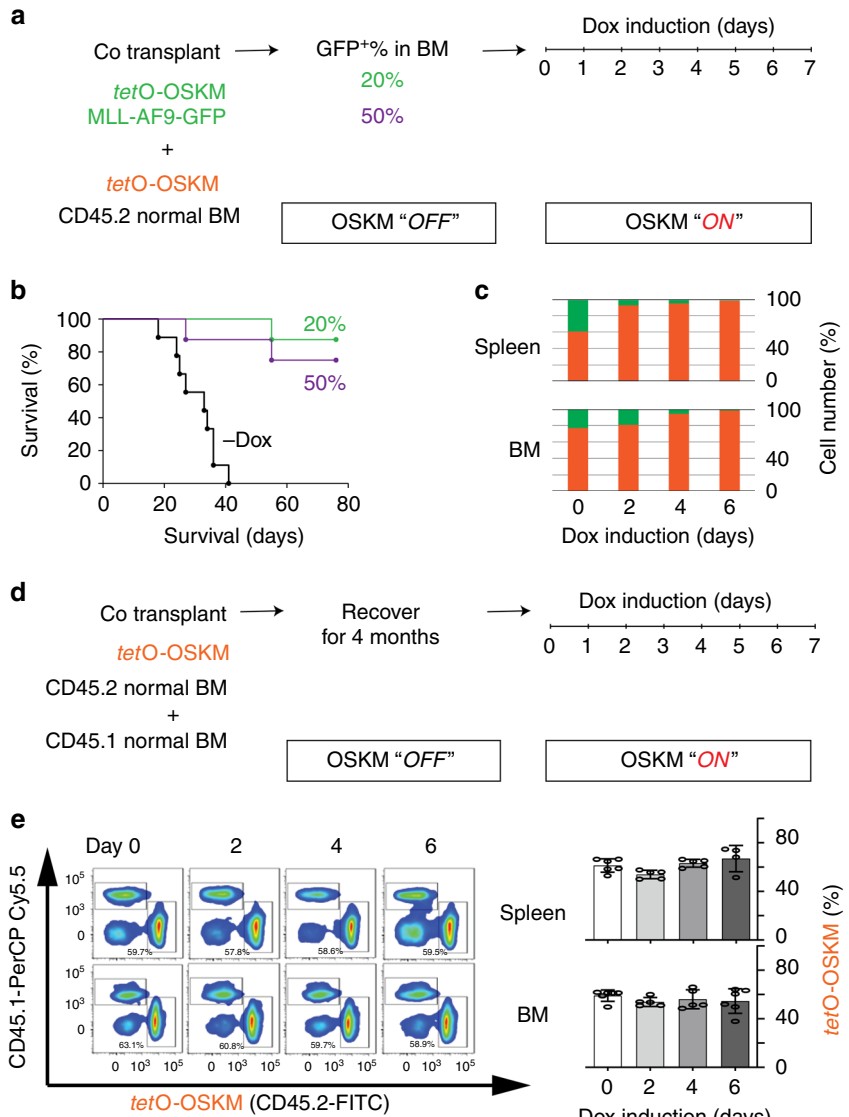

**Fig. 2 Normal bone marrow cells are insensitive to OSKM. a** Schematic of the competitive transplantation assay. Briefly, $2 \times 10^5$ MLL-AF9-OSKM cells and $2 \times 10^7$ *tet*O-OSKM CD45.2$^+$ cells were co-transplanted into lethally irradiated (9.5 Gy) CD45.1 recipient mice. When MLL-AF9-OSKM reached the indicated percentage in the bone marrow of the recipient mice, Dox treatment was given as described in Fig. 1a. BM, bone marrow. **b** Left: Survival curves of CD45.1 recipient mice after Dox withdrawal. Black trace is the no Dox treatment control. Green and purple traces are the survival curves for CD45.1 recipient mice whose MLL-AF9-OSKM cells reached 20% and 50% of bone marrow, respectively, before Dox treatment ($n = 9$–10). **c** Relative percentage of GFP$^-$CD45.2$^+$ normal cells (orange) and GFP$^+$CD45.2$^+$ AML cells (green) in the BM of CD45.1$^+$ recipient mice that underwent Dox treatment ($n = 4$). **d** Schematic of the competitive transplantation assay. Briefly, *tet*O-OSKM CD45.2$^+$ BM cells together with CD45.1$^+$ cells were transplanted into lethally irradiated CD45.1 recipient mice. After 4 months recovery time, CD45.1 recipient mice were given Dox treatment as described in Fig. 1a. **e** Left: FACS analysis of spleen (upper) and bone marrow cells (lower) from CD45.1 recipient mice. Dox treatment was given as described in Fig. 1a; Right: in vivo kinetics of *tet*O-OSKM CD45.2$^+$ cells in spleen and bone marrow ($n = 4$–6, two independent experiments). One-way ANOVA. Error bars show SEM.

withdrawal (Fig. 2b). The amount of MLL-AF9-OSKM cells in BM dropped from 24% to almost zero, which is consistent with our previous observations (Fig. 1c). The amount of *tet*O-OSKM CD45.2$^+$ cells, on the other hand, increased (Fig. 2c). To further confirm ectopic expression of OSKM factors has little effect on non-leukemia cells, we employed the reconstitution model assay[22]. In this assay, we co-injected whole BM cells from CD45.1$^+$ and *tet*O-OSKM CD45.2$^+$ transgenic mice into lethally irradiated CD45.1$^+$ recipient mice (Fig. 2d). After 4 months reconstitution time, we started 7 days of Dox treatment and monitored the fate of CD45.2$^+$ cells in BM and spleen of the recipient mice using flow cytometry. Through the 7-day Dox treatment, the number of CD45.2$^+$ cells remained similar

(Fig. 2e) and no alteration of their differentiation potential was observed (Supplementary Fig. 2b). More importantly, transient OSKM expression did not reduce the fraction of LKS$^+$ HSPCs and LKS$^-$ HPCs in CD45.2$^+$ cells (Supplementary Fig. 2c). Furthermore, at the end of Dox treatment, CD45.2$^+$ cells showed similar proliferation and differentiation patterns as cells isolated from a no Dox control (Supplementary Fig. 2d).

Finally, we examined the reprogramming efficiencies of *tet*O-OSKM CD45.2$^+$ and MLL-AF9-OSKM cells. As cells at different stages of the differentiation hierarchy might have different reprogramming efficiencies, we investigated three different types of *tet*O-OSKM CD45.2$^+$ cells and MLL-AF9-OSKM cells that represented the stem cell stage, progenitor cell stage, and fully

committed cell stage. Notably, HSPCs and HPCs exhibited high reprogramming efficiency by ectopic OSKM induction, but granulocytes and leukemia cells did not, especially LSCs (Supplementary Fig. 2e). Taken together, these results indicate that 7 days of ectopic expression of OSKM factors neither reduces the number of tetO-OSKM CD45.2$^+$ cells nor affects the differentiation potential of tetO-OSKM CD45.2$^+$ cells. Furthermore, the cell reduction observed in response to expression of OSKM factors is specific to leukemia cells.

In addition, we transduced cord blood (CB) CD34$^+$ cells and THP-1 cell line with OSKM lentiviruses and also observed that OSKM induction had only a mild effect on CD34$^+$ cells but caused a dramatic reduction in cell number along with impaired colony-forming ability (Supplementary Fig. 3a, b).

**Immune cells are not major killing force upon OSKM elevation.** To investigate the potential cause of the eradication of leukemia cells, we first analyzed transcriptomic changes by microarray. MLL-AF9-OSKM cells were grown until they reached 50% of BM in recipient mice before administering Dox-containing water. After 48 h or 72 h of Dox treatment, we generated cDNA libraries from MLL-AF9-OSKM cells for microarray analysis (Fig. 3a). Compared with a no Dox control, eight out of the top 15 most enriched biological pathways were associated with immune response. We therefore tested if the immune response is the main driver of the loss of leukemia cells when OSKM factors are induced. To provide functional evidence, we transplanted MLL-AF9-OSKM cells into sub-lethally irradiated NOD/SCID mice[23] that lack T and B cells. When MLL-AF9-OSKM cells reached 50% of BM in NOD/SCID recipient mice, anti-CD122 antibody, which blocks differentiation of T and Natural Killer (NK) cells, was injected intraperitoneally, followed by Dox treatment (Fig. 3b). Similar to what we observed previously (Fig. 1), without Dox, all NOD/SCID recipient mice died, regardless of anti-CD122 treatment. Conversely, after Dox treatment, over 80% of NOD/SCID recipient mice survived (Fig. 3c). Although NOD/SCID mice lack T and B cells, they still contain macrophages, which have been shown to kill cancer cells via phagocytosis[24]. To further assess the effect of macrophages on the selective killing we observed, we removed macrophages in NOD/SCID recipient mice via clodronate liposome (CL) treatment[24,25] before Dox induction (Fig. 3b). Consistent with previous studies, the level of macrophages in the BM and spleen of the NOD/SCID recipient mice was drastically reduced after CL treatment (Supplementary Fig. 4a). Macrophage depletion had no effect on the survival rate of the NOD/SCID recipient mice (Fig. 3d), nor did it change the percentage of MLL-AF9-OSKM cells in peripheral blood after Dox treatment (Supplementary Fig. 4b). These data indicated that the selective killing of MLL-AF9-OSKM cells was not dependent on T, B, NK cells, or macrophages, suggesting that the immune response is not a major driver of selective leukemia cell depletion.

**OSKM elevation leads to apoptosis of AML cells.** Because the microarray experiment was done using in vivo Dox-treated MLL-AF9-OSKM cells, many secondary factors could have been introduced. To more directly assess the effect of OSKM expression, we treated cells in vitro with Dox before examining transcriptomic changes. Moreover, as an immune response could be a later reaction to how MLL-AF9-OSKM responded to Dox treatment, we assessed earlier time points (0, 3, 6, 12, and 24 h of Dox treatment). Hierarchical clustering of RNA-seq data from MLL-AF9-OSKM cells showed a tight correlation between biological replicates as well as a separation between early and late time points of Dox induction (Supplementary Fig. 5a), suggesting that

the effect of OSKM expression at the transcriptome level is modest at 3 h, but then becomes the dominant factor in transcriptome change. TetO-OSKM cKit$^+$ cells collected at 3 h and 6 h post-induction clustered together with the no Dox treatment control (Supplementary Fig. 5b). After 12 h of Dox treatment, OSKM expression contributed more substantially to transcriptomic changes of tetO-OSKM cKit$^+$ cells. These results suggest that cancerous and non-cancerous cells respond to OSKM induction on different time scales. To investigate direct effects of OSKM, we focused on those differentially expressed genes that contain OSKM binding motifs in their promoters and then performed GO term analysis (Fig. 4a). The top 20 enriched biological processes included positive regulation of cell death and immune response. Given our previous results excluding the immune response, we focused on cell death as the possible cause for the selective eradication of cancer cells.

The positive regulation of cell death could come from necrosis or apoptosis. We observed that the longer into the Dox treatment, the more MLL-AF9-OSKM cells were Annexin V$^+$ 7AAD$^-$ or Annexin V$^+$ 7AAD$^+$ (Fig. 4b). Very few or no cells were 7AAD$^+$ and Annexin V$^-$, and we did not observe any necrotic cells by electron microscopy (Supplementary Fig. 5c), indicating that necrosis is not the cause of cell death. We then examined the protein level of p53, PUMA, and cleaved Caspase-3, all of which are involved in the major molecular pathways that lead to apoptosis[26,27]. All three showed increased protein levels (Fig. 4c). As tetO-OSKM cKit$^+$ cells also showed increased expression of genes involved in the positive regulation of cell death (Fig. 4a), we examined Annexin V and 7AAD levels in these cells as well (Supplementary Fig. 5d). In contrast to MLL-AF9-OSKM, only a very small percentage of tetO-OSKM cKit$^+$ cells were Annexin V$^+$. The expression levels of the genes in the positive regulation of cell death category in tetO-OSKM cKit$^+$ cells went up and then back down (Fig. 4a), which is consistent with the Annexin V protein levels (Supplementary Fig. 5d). We therefore attribute this early fluctuation to the in vitro culture condition, rather than Dox treatment. Together, these data indicated that these MLL-AF9-OSKM cells were undergoing apoptosis during Dox treatment and transient OSKM induction did not cause apoptosis in tetO-OSKM cKit$^+$ cells. In addition, human THP-1 cells had a much higher apoptotic level than cord blood CD34$^+$ cells after OSKM induction (Supplementary Fig. 3c).

To measure the real-time apoptosis in MLL-AF9-OSKM cells after Dox treatment, we took advantage of a switch-on, fluorescence-based DEVDase activity indicator (C3AI)[28]. This C3AI indicator contains a caspase-3 cleavage site as a switch-on console. When apoptosis occurs, caspase-3 or caspase-3-like proteases cleave the console, releasing fluorescence and C3AI rapidly becomes cerulean (cyan fluorescent protein, CFP). We transfected MLL-AF9-OSKM cells with the C3AI indicator plasmid. After 48 h of incubation, we performed CFC assay with the addition of puromycin for 10 days and picked out single colonies to ensure 100% transfection efficiency. These single colonies were then expanded before being split into two groups: with or without Dox treatment for 24 h, followed by morphology monitoring using confocal laser scanning microscopy. Consistent with the elevated protein level of Caspase-3, fluorescence was observed when MLL-AF9-OSKM cells were treated with Dox for 24 h (Fig. 4d), indicating that cells were undergoing apoptosis in vivo. We monitored the fluorescence from 0 to 24 h Dox treatment to determine when Caspase-3 started to function (Fig. 4e). Although the reduction in MLL-AF9-OSKM cells started 1 day after Dox treatment (Fig. 1c), fluorescence arising from Caspase-3 function, indicating activation of apoptosis, started as early as 14 h.

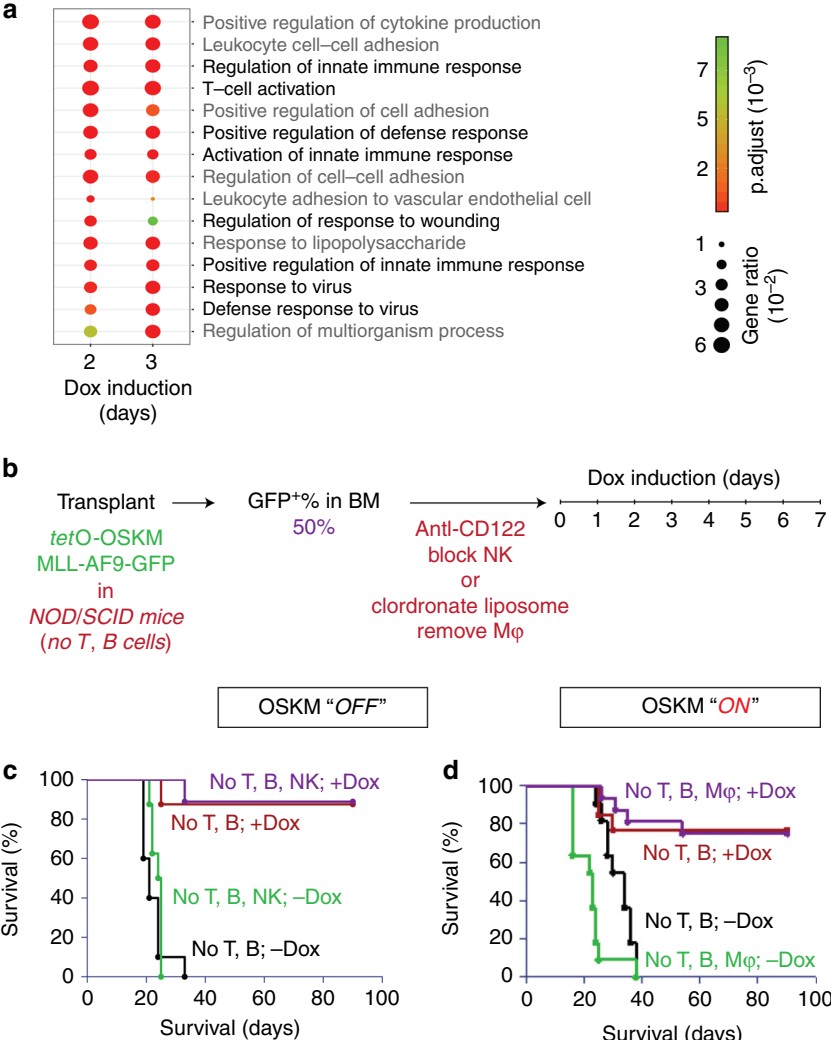

**Fig. 3 Immune response is not a major driver of selective leukemia cell depletion. a** Gene Ontology (GO) analysis on differentially expressed genes. Dot plot of the top 15 enriched biological pathways. The size of circle indicates the percentage of differentially expressed genes within the particular biological pathway. Black font indicates the biological pathways that are related to immune response; gray font indicates other biological pathways. **b** Schematic of the experimental procedure. Briefly, MLL-AF9-OSKM cells were transplanted into sub-lethally irradiated (2.0 Gy) NOD/SCID mice that lack T and B cells. When MLL-AF9-OSKM cells reached 50% of bone marrow of NOD/SCID recipient mice, 200 μg of purified anti-CD122 antibody or clodronate liposome (CL) was intraperitoneally injected, followed by Dox treatment as described in Fig. 1a. BM, bone marrow. **c** Survival curves of the NOD/SCID recipient mice that had anti-CD122 antibody treatment after Dox withdrawal. Green and black traces are the no Dox treatment control for the NOD/SCID recipient mice that received or not received anti-CD122 antibody, respectively. Purple and red traces are the NOD/SCID recipient mice that received or not received anti-CD122 antibody, respectively, prior to Dox treatment ($n = 8$–11). **d** Survival curves of the NOD/SCID recipient mice that CL treatment after Dox withdrawal. Green and black traces are the no Dox treatment control for the NOD/SCID recipient mice that received or not received CL, respectively. Purple and brown traces are the NOD/SCID recipient mice that received or not received CL, respectively, prior to Dox treatment ($n = 8$–11).

**OSKM elevation alters chromatin accessibility in AML cells.** *Oct4*, *Sox2*, and *Klf4* are all pioneer factors that can bind to closed chromatin and recruit other transcription factors to regulate target gene expression[29]. *c-Myc*, albeit not a pioneer factor, has been shown to facilitate OSK binding to chromatin[30]. During the reprogramming process, the chromatin state at lineage-specific transcription factor loci changes from open to closed, accompanied by downregulation of these factors, followed by a change from closed to open at pluripotency gene loci, concurrent with upregulation of these genes[4]. We therefore tested if cancer cells that received ectopic expression of OSKM factors adapted similar process.

To examine chromatin accessibility change during OSKM induction, we applied ATAC-seq in MLL-AF9-OSKM cells and *tet*O-OSKM cKit$^+$ cells that went through 0, 3, 6, 12 or 24 h of

Dox induction. Regions showing differences in openness (FDR < 0.05) compared to the no Dox control counterpart were defined as either closed to open (CO), if the openness was greater after OSKM induction, or open to closed (OC), if the openness was less after OSKM induction. In *tet*O-OSKM cKit$^+$ cells, similar to the transcriptome changes, we did not observe a chromatin change until 12 h of induction (Right panel, Fig. 4f and Supplementary Fig. 6a). At 12 and 24 h of induction, more OC regions were observed. As expected from the first step of reprogramming, these OC regions contained loci of hematopoietic-specific lineage transcription factors, such as PU.1 and Myb (Supplementary Fig. 6b). Among those CO regions, *Oct4*, but not *Sox2*, *Klf4*, or *Nanog*, started to open up after 3 h of induction (Supplementary Fig. 6c). This is not surprising as these pluripotency loci would not show chromatin being opened up until at least 7 days of

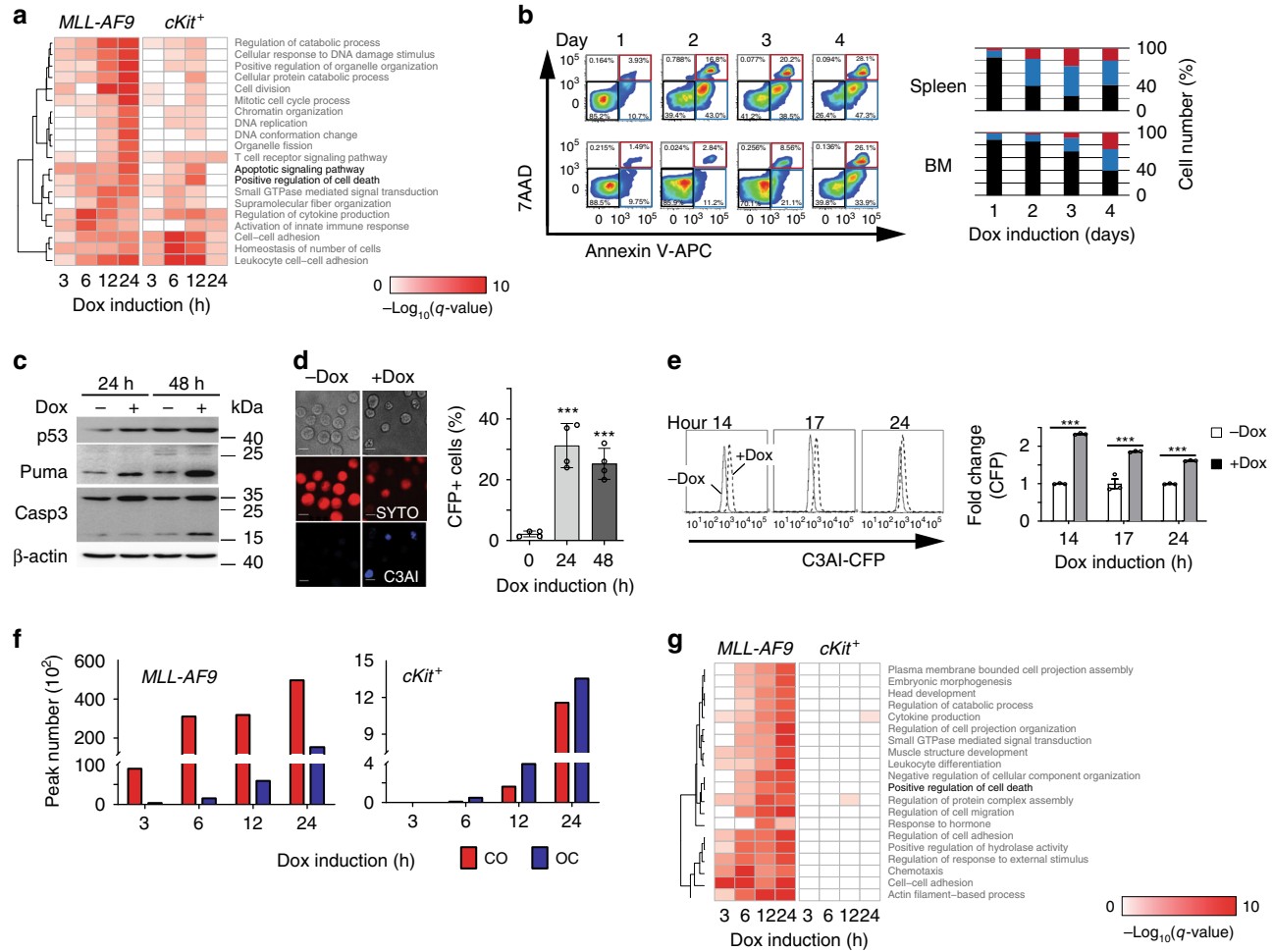

**Fig. 4 Mechanism of OSKM dependent leukemia cell apoptosis. a** Gene Ontology (GO) analysis of differentially expressed genes that contain OSKM motifs inside promoter regions for MLL-AF9 cells (left panel) and cKit[+] cells (right panel). Heatmap of the top 20 enriched biological pathways. Black font indicates the biological pathways that are related to cell death; gray font indicates other biological pathways. **b** Left: Flow plots show the apoptosis rate of MLL-AF9-OSKM cells from spleen (upper) and bone marrow cells (lower) of recipient mice that underwent Dox treatment; Right: Bar-graph plots summarize the populations of MLL-AF9-OSKM cells in spleen (upper) and bone marrow (lower). Gray box and gray bar show 7AAD[+]/Annexin V[−] population; black box and black bar show 7AAD[−]/Annexin V[−] population; red box and red bar show 7AAD[+]/Annexin V[+] population; blue box and blue bar show 7AAD[−]/Annexin V[+] population (n = 3–5, 2 independent experiments). BM, bone marrow. **c** Western blot of p53, Puma, and Caspase 3 from MLL-AF9-OSKM cells that underwent Dox treatment compared to no Dox. β-actin was used as internal control. **d** Left: Upper panels represent C3AI cells with or without 24 h of Dox treatment. Middle panels show the location of the nuclei of C3AI cells using SYTO red fluorescent dye. Lower panels show the presence of fluorescence (CFP) from C3AI cells undergoing apoptosis. Scale bars: 10 μm. Right: Quantification of the percentage of CFP[+] cells from C3AI cells that underwent Dox treatment. 200 plus cells per view were counted, with four replicates per group. Two independent experiments. ***$p < 0.001$, One-way ANOVA. Error bars show SEM. **e** Left: Flow histogram displaying the fluorescence level of C3AI cells treated with Dox for the indicated time. Right: Bar graphs of the fluorescence intensity of C3AI cells treated with Dox for the indicated time The fluorescence intensity was normalized to C3AI cells without Dox treatment (n = 3, 3 independent experiments). ***$p < 0.001$, two-tailed Student's t test. Error bars show SEM. **f** Bar graphs of chromatin differential openness regions in tetO-OSKM AML cells (left) and cKit[+] cells (right) that underwent Dox treatment. Red bar shows the amount of regions where chromatin is more open than 0 h control, called as close to open (CO). Blue bar shows the amount of regions where chromatin is less open than 0 h control, called as open to close (OC). **g** Gene Ontology (GO) analysis of chromatin differential openness regions. Heatmap of the top 20 enriched biological pathways.

induction[4]. MLL-AF9-OSKM cells, on the other hand, behaved very differently (Left panel, Fig. 4f and Supplementary Fig. 6a). Not only were most of the chromatin changes in CO regions, but the OC regions were not observed until after 12 h of induction. Furthermore, the changes in chromatin openness regions were 25–40 fold greater in MLL-AF9-OSKM cells than tetO-OSKM cKit[+] cells.

To understand the functional differences arising from these changes in the chromatin landscape, we looked genome-wide at the location of changed chromatin openness regions and at the genes they are associated with. Differentially opened chromatin regions that fell within a 2 kb window centered around TSSs were

assigned to that particular TSS. After annotation, we examined which biological processes were enriched in these differentially opened chromatin regions (Fig. 4g). Similar to what we observed at the transcriptomic level (Fig. 4a), the chromatin dynamics in regions containing genes associated with positive regulation of cell death were changed in MLL-AF9-OSKM cells, but not in cKit[+] tetO-OSKM cells. Interestingly, even though tetO-OSKM cKit[+] cells started showing transcriptome changes as early as 3 h post-induction, no changes to chromatin openness were detectable at this time point. However, when OSKM expression started to play a major role in driving the transcriptomic changes, we observed a concomitant change in chromatin openness. Similarly,

in MLL-AF9-OSKM cells, expression of OSKM factors started to have an effect on the transcriptome as early as 3 h of induction, and we also observed a significant chromatin openness change at that time, indicating that the temporal dynamics of transcriptomic changes are coordinated with chromatin changes.

**Sox2 and Klf4 are largely responsible for killing AML cells.** To identify transcriptional factors involved in the selective depletion of MLL-AF9-OSKM cells in response to OSKM factor induction, we applied the Homer bioinformatic tool to systematically identify motifs within regions of differential chromatin openness (Fig. 5a, with complete set of significant motifs in Supplementary Data 1). The top 20 enriched motifs included binding sites for the hematopoiesis associated ETS and RUNT family transcription factors as well as the *Sox* and *Klf* family of transcription factors. Surprisingly, *c-Myc* motifs were not enriched in differential chromatin openness regions at any of the tested time points, whereas *Oct4* motifs were observed only in *tet*O-OSKM cKit+ cells after 24 h of induction.

c-Myc overexpression is known to induce apoptosis in cancer cells and reprogramming[31–33], but it did not show up in our motif search in the ATAC-seq dataset. This made us wonder if all reprogramming factors are required for the selective killing of cancer cells that we observed. To dissect the contribution of each reprogramming factor, we measured the growth curve of MLL-AF9 and cKit+ BM cells transfected with retrovirus that contained *Oct4*, *Sox2*, *Klf4*, and *c-Myc* individually or in different combinations. After 48 h of transfection, we confirmed the overexpression of individual OSKM genes (Supplementary Fig. 7a). As expected, without any reprogramming factors, the amount of MLL-AF9 cells increased (Fig. 5b). To our surprise, either *c-Myc* or *Oct4* overexpression alone showed similar increases in cell number as seen without any reprogramming factors. Thus, *c-Myc* and *Oct4* alone cannot kill MLL-AF9 cells. Conversely, the amount of MLL-AF9 cells was dramatically reduced when either *Sox2* or *Klf4* alone were overexpressed. This reduction was even greater when both *Sox2* and *Klf4* were overexpressed together. Regardless of the composition of the reprogramming factor cocktail, the amount of cKit+ BM cells steadily increased (Fig. 5b). It is worth noting that the proliferation rate of cKit+ cells was modestly reduced when *Sox2* and *Klf4* were co-overexpressed.

After 4 days of transfection, we measured the percentage of Annexin V+ cells among each test group (Supplementary Fig. 7b). MLL-AF9 cells overexpressing *Sox2*, *Klf4*, or *Sox2/Klf4* displayed a high fraction of Annexin V+ cells. MLL-AF9 cells overexpressing *Oct4* or *c-Myc*, on the other hand, displayed a similar fraction of Annexin V+ cells as the empty control. Regardless of the composition of reprogramming factors, cKit+ BM cells displayed slightly but significantly fewer Annexin V+ cells than the empty control. Furthermore, colony assay demonstrated that overexpression of *Sox2* or *Klf4* alone significantly reduced the clonal growth of leukemia cells in vitro (Supplementary Fig. 7c). Inhibitory growth, along with increased apoptosis by *Sox2* and *Klf4*, was also observed in the THP-1 cells but not significantly in human cord blood CD34+ cells (Fig. 5c and Supplementary Fig. 7d, e). *Sox2* and *Klf4*, instead of *c-Myc*, appear to play a major role in the elimination of leukemia cells, at least for the leukemia type tested in our study.

**H3K9 methylation inhibitor selectively inhibits AML cells.** Chromatin alteration is usually accompanied with histone modification changes. To explore this, we used Western blotting to examine the dynamics of histone marks at early time points following Dox induction. Among all tested histone modifications,

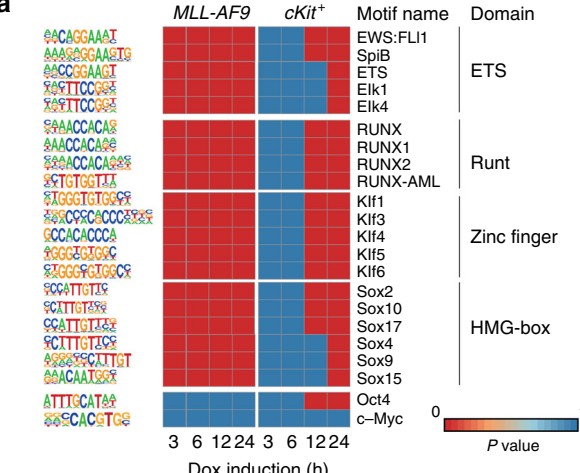

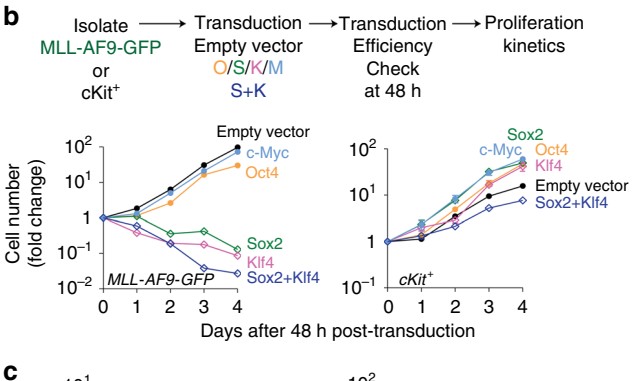

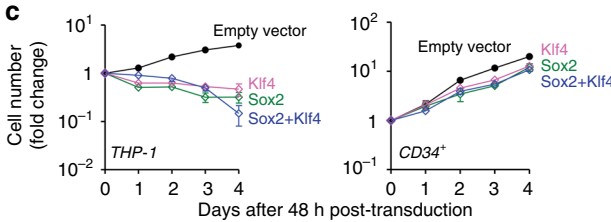

**Fig. 5 Sox2 and Klf4 are responsible for leukemia cell killing activity.** **a** Heatmap of the top 20 significant motifs enriched from chromatin differential openness regions of AML and cKit+ OSKM cells that underwent Dox treatment. Oct4 and c-Myc motifs are plotted. **b** Upper: Schematic of the in vitro liquid culture assay. Briefly, AML and cKit+ cells were transfected with the indicated reprogramming factor cocktail, followed by efficiency check at 48 h post-transduction before growth curves were monitored daily. The cell number was normalized to Day 0 before plotting. Lower left: Growth curve of AML cells that contained different reprogramming factor cocktails. Lower right: Growth curve of cKit+ cells that contained different reprogramming factor cocktails. Black trace is the empty vector control. Light blue, orange, green, and pink traces are cells transfected with individual c-Myc, Oct4, Sox2, and Klf4 factors, respectively. Dark blue trace is cells transfected with both Sox2 and Klf4 (n = 3, 3 independent experiments). Error bars show SEM. **c** In vitro liquid culture assays for THP-1 cells (left) and normal CB CD34+ cells (right) after OSKM ectopic overexpression. The cell number at day 0 was normalized to 1. The relative cell number at different time points is shown (n = 3, three independent experiments). Error bars show SEM.

only H3K9me3 was downregulated in leukemia cells and not in *tet*O-OSKM cKit+ cells (Fig. 6a and Supplementary Fig. 8a). It is worth noting that H3K9 levels were also decreased in MLL-AF9 cells that harbored *Sox2 + Klf4* plasmid (Supplementary Fig. 8b). To functionally explore whether H3K9 demethylation selectively

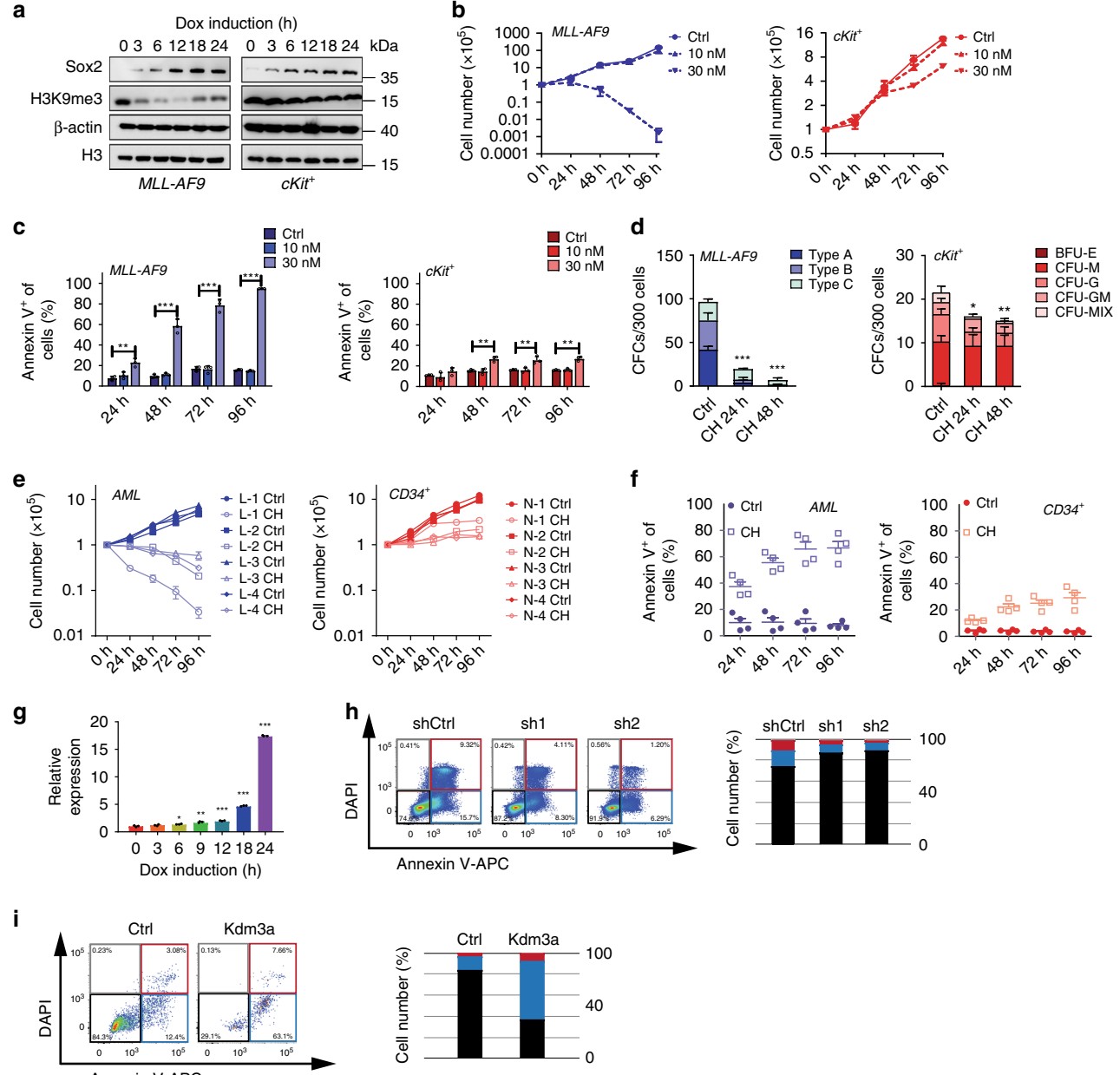

**Fig. 6 Inhibition of H3K9me3 induces apoptosis of leukemia cells. a** Western blots of Sox2, H3K9me3, H3, and β-actin in MLL-AF9-OSKM AML cells and *tet*O-OSKM cKit+ bone marrow cells after Dox induction at different time points. Representative graphs of three independent experiments are shown. *$p <$ 0.05, **$p < 0.01$, ***$p < 0.001$, one-way ANOVA. Error bars show SEM. **b** Cell counts of murine AML cells and cKit+ bone marrow cells after treatment with the indicated concentration of chaetocin or DMSO. Two independent experiments. **c** Apoptosis of murine AML cells and cKit+ BM cells after treatment with the indicated concentration of chaetocin or DMSO for 24, 48, 72 and 96 h. Three independent experiments. **$p < 0.01$, ***$p < 0.001$, one-way ANOVA. **d** Colony formation of murine AML cells and cKit+ BM cells after treatment of chaetocin (30 nM) for 24 or 48 h. CH, chaetocin. Three independent experiments. *$p < 0.05$, **$p < 0.01$, ***$p < 0.001$, one-way ANOVA. **e** Cell counts of human AML and normal CD34+ cells after treatment with chaetocin (30 nM) or DMSO for 24, 48, 72, 96 h. Two independent experiments with four individual samples. **f** Apoptosis of human AML and normal CD34+ cells after treatment with chaetocin (30 nM) or DMSO for 24, 48, 72, 96 h. Two independent experiments with 4 individual samples. One-way ANOVA. **g** qRT-PCR detection of *Kdm3a* in MLL-AF9-OSKM AML cells and *tet*O-OSKM cKit+ BM cells after 3–24 h treatment with Dox in vitro ($n = 3$, two independent experiments). **h** Flow plots and histograms show the apoptotic rate of MLL-AF9-OSKM cells transduced with *Kdm3a* shRNA. The data were taken 48 h after Dox treatment ($n = 3$–5, two independent experiments). **i** Flow plots and histograms show the apoptotic rate of cKit+ bone marrow cells overexpressing *Kdm3a*. The data were taken 48 h after transduction ($n = 5$, two independent experiments). All error bars show SEM.

affects leukemia cells, we treated AML cells and cKit+ BM cells with chaetocin, a small-molecule inhibitor of H3K9 methylation[34–36] (Supplementary Fig. 8c). After treatment, especially at 30 nM of chaetocin, the number of viable AML cells was significantly reduced, whereas the growth of cKit+ BM cells was mildly affected (Fig. 6b). Chaetocin treatment led to increased

apoptosis in AML cells, while only slightly increasing apoptosis in normal cKit+ cells (Fig. 6c). Moreover, 30 nM of chaetocin led to loss of clonogenic potential and induction of differentiation in AML cells (Fig. 6d). However, cKit+ BM cells showed only a modest reduction in colony-forming ability (Fig. 6d). Importantly, similar results were also obtained in human samples that

received this treatment. When human AML CD34$^+$ cells and normal CD34$^+$ cells were treated with 30 nM of chaetocin, a selective anti-proliferative effect on the primary human AML cells was observed (Fig. 6e, f). In addition, the colony-forming ability of the AML cells was more significantly inhibited than was that of the normal CD34$^+$ cells (Supplementary Fig. 8d, e).

To identify the cause of OSKM-induced H3K9 dysregulation, we systematically examined the transcriptome data for differentially expressed H3K9 demethylases and methyltransferases. *Kdm3a* and *Suv39h1/Suv39h2*, but not Setdb1, showed significant dysregulation, although the change in Suv39h1/Suv39h2 occurred later than the appearance of H3K9 dysregulation (Fig. 6a and Supplementary Fig. 9a–d). We therefore focused on *Kdm3a*, a H3K9 demethylase[37]. Using qPCR validation, we confirmed that the expression of *Kdm3a* was progressively upregulated in Dox-treated leukemia cells but not in normal HSPCs (Fig. 6g and Supplementary Fig. 9a, b). Importantly, the OSKM-induced apoptosis in MLL-AF9-OSKM cells could be rescued by knockdown of *Kdm3a* (Fig. 6h and Supplementary Fig. 9e, f). In addition, overexpression of *Kdm3a* in normal cKit$^+$ HSPCs led to apoptosis and caused cell death (Fig. 6i and Supplementary Fig. 9g). Taken together, H3K9 methylation activity appears to be essential for the survival of AML cells but less critical for that of normal hematopoietic cells, and *Kdm3a* demethylase is involved in OSKM-induced apoptosis.

## Discussion

Despite the known shared mechanisms between iPSC reprogramming process and cancer development, the current study demonstrates an anti-cancer effect of reprogramming factors on established cancer cells. With the leukemia models in which OSKM factors could be conditionally controlled by Dox, this study reveals an acute effect of the reprogramming stress on cancer cells. This process is apparently either independent of or apart from the de-differentiation effects caused by the OSKM factors. Although *c-Myc* overexpression is known to induce apoptosis under certain circumstances[31,32,38], elevation of *c-Myc* itself is not sufficient to induce the killing effect in the leukemia model used in our study whereas *Sox2* or *Klf4* alone, or the synergistic effect when both factors are combined, is sufficient to selectively eliminate MLL-AF9-OSKM cells. Interestingly, LSCs are more sensitive to OSKM than other leukemia cell populations. In contrast, normal HSPCs are insensitive to short-term OSKM induction. Notably, the OSKM-induced apoptotic effect was also found to affect two other types of leukemia cells, i.e., MLL-NRIP3 and Notch1-induced leukemia cells, indicating a general phenomenon capable of affecting not only different types of leukemia but also various solid tumors.

The finding regarding the less sensitive of normal HSPCs to OSKM increases the potential usefulness of the OSKM approach in developing new therapeutic agents against cancer. In addition, the genomic instability that is commonly associated with cancer cells and not with normal cells may account for the increased cell death upon reprogramming. This possibility may hold the truth for certain types of cancer. However, the MLL-AF9 leukemia cells used in our study have been shown to be genomically stable[39]. As reprogramming is known to cause DNA damage in these cells[40], more detailed analyses with regard to DNA damage response dependent or independent of p53 activation[41] may provide additional insight.

Different epigenetic states lead to diverse cell fates. Previous studies have demonstrated that a reduction of H3K9 methylation yields an accessible chromatin state and facilitates reprogramming[42–45]. Interestingly, leukemia cells appeared to respond differently to this epigenetic trigger. When OSKM factors were

induced in vivo, *Kdm3a*, a canonical H3K9 demethylase, was upregulated and global H3K9me3 levels were reduced. This change was accompanied by upregulation of apoptotic regulators. Inhibition of *Kdm3a* in MLL-AF9 AML cells that underwent OSKM induction could rescue cell number reduction. Furthermore, MLL-AF9 AML cells treated with a small-molecule H3K9 methylation inhibitor could mimic the OSKM-induced selective eradication effect. This suggests H3K9 methylation may be an attractive epigenetic target for the specific induction of cell death in leukemia cells. As the innate epigenetic status could affect OSKM targets, ultimately resulting in different cell fates between normal cKit$^+$ HSPCs and MLL-AF9 AML cells, future studies should focus on the role of pre-existing chromatin status in promoting this selective eradication effect.

Our study demonstrates the differential effects of reprogramming factors on cancer initiation and maintenance. Because oncogenesis is considered a de-differentiation process, one would anticipate that cancer cells may be easily reprogrammed into iPSCs by OSKM factors or by other means, such as nuclear transfer. Paradoxically, cancer cells have been shown to be difficult to reprogram into a pluripotent state[46]; however, some types of cancer cells have been shown to be reprogrammable although at an extremely low efficiency[12,13,47–51]. Meanwhile, it was reported that reprogramming not only inhibited malignant features in colon cancer but also restored the sensitivity of the leukemia cells to imatinib[15,51]. Apart from reprogramming, in our own analysis, only non-LSCs could be reprogrammed into iPSCs, and the majority of leukemia cells underwent apoptosis upon the same reprogramming treatment (Supplementary Fig. 2e). More surprisingly, the LSCs were even more sensitive to OSKM. This result reinforces the idea that reprogramming potential toward a pluripotent state is not necessarily correlated with differentiation stage; we documented this concept in a previous study, in which HSCs were less efficiently reprogrammed than both HPCs and granulocytes via a nuclear transfer system in mice[52]. Indeed, the selective killing effect on leukemia cells also reflects an ancient Chinese philosophy, Wu Ji Bi Fan (extremes meet), meaning that when things reach an extreme, they can only move in the opposite direction. Therefore, the basic principles on effective reprograming apart cancer elimination (ERACE) established in this study will enable our future efforts to develop more effective therapeutics, particularly new epigenetic agents, against cancer.

## Methods

**Mice**. B6-Ly5.1, B6-Ly5.2 and NOD/SCID mice were purchased from the animal facility of the State Key Laboratory of Experimental Hematology (SKLEH, Tianjin, China). The OSKM mice were gifts from Dr. Shaorong Gao (Tongji University, Shanghai, China). All animal procedures were done in compliance with the animal care guidelines approved by the Institutional Animal Care and Use Committees of the SKLEH and the Institute of Hematology.

**Cell Lines**. 293T and THP-1 cells lines were obtained from SKLEH's experimental pathology cell bank. All the cells were authenticated by examination of morphology and growth characteristics, and were confirmed to be mycoplasma free.

**Plasmids and virus production**. MSCV-MLL/AF9-PGK-PURO was generously provided by Dr Chi Wai So. The PGK-PURO segment was replaced by IRES-green fluorescent protein (GFP) to form the MSCV-MLL/AF9-IRES-GFP construct. The CMV-CC3AI⁻PURO plasmid for the Cerulean-based caspase-3-like protease activity indicator (CC3AI) apoptosis reporter system was obtained from Dr. Binghui Li. The OSKM vectors pRSC-SFFV-Oct4/Flag-PGK-YFP, pRSC-SFFV-Klf4/Flag-PGK-GFP, pRSC-SFFV-c-Myc/Flag-PGK-YFP, pRSC-SFFV-Sox2-PGK-RFP, LV-tetO-Sox2-mCherry and LV-tetO-Klf4-EGFP were kindly provided by Dr. Xiaobing Zhang (Loma Linda University, USA). The LV-shRNA-RFP and LV-cDNA-RFP of *Kdm3a* lentiviruses were produced by GeneChem (Shanghai, China). For retrovirus production, the target plasmid, together with pKat and pVSVG, was transfected into the 293T cell line using Lipofectamine 2000. The supernatant was harvested after 48 and 72 h of culture and concentrated using an Amicon filter. For lentiviral production, the target plasmid was transfected together with pSPAX2 and pMD2G.

**Leukemia mouse model**. Fresh whole BM cells were harvested and enriched using lineage cell depletion beads (Miltenyi). Lin⁻ stem and progenitor cells were incubated overnight in Iscove's modified Dulbecco's medium (IMDM) with 15% fetal bovine serum (FBS), 50 ng/mL mSCF, 10 ng/mL mIL-3 and 10 ng/mL mIL-6 to promote cell cycle entry. The prestimulated cells ($5 \times 10^5$) were then spinoculated with retroviral supernatant in the presence of 6 µg/mL polybrene (Sigma) for 90 min at 650 xg. After 2 days of culture, $5 \times 10^5$ transduced cells were injected into lethally irradiated mice (9.5 Gy)[53].

**Doxycycline treatment**. Dox (1 mg/mL, Sigma) was administered to the mice in their drinking water, which was supplemented with 7.5% sucrose, for 7 days. For in vitro cell cultures, Dox was used at a concentration of 2 µg/ml.

**Flow cytometry**. For cell sorting experiments using mouse HSPCs, cKit⁺ cells were enriched before flow cytometry using cKit magnetic beads (Miltenyi Biotec). Subsequently, the cells were stained with a lineage cocktail, cKit (eBioscience, 17-1171-82, 1:200) and Sca-1 (eBioscience, 25-5981-82, 1:200) antibodies. The lineage cocktail include Gr-1 (Biolegend, 108424, 1:400), Mac-1 (Biolegend, 101226, 1:400), B220 (Biolegend, 103224, 1:400), CD4 (Biolegend, 100414, 1:400), CD8 (Biolegend, 100714, 1:400), CD3 (Biolegend, 100330, 1:400) and Ter-119 (Biolegend, 116223, 1:400). DAPI (1 mg/mL, Sigma-Aldrich) was used to exclude dead cells. For LSC analysis, nucleated BM cells were stained with lineage-specific antibodies, Sca-1 and cKit antibodies. The lineage-specific antibodies include CD3, CD4, CD8, B220, Gr-1, Ter119 and CD127 (Biolegend, 135040, 1:400). For apoptotic analysis, the cells were stained with Annexin V and 7-AAD according to the manufacturer's recommendations (BD Biosciences, 550475, 1:100). A modified LSR II flow cytometer with four lasers (355 nm, 488 nm, 561 nm, and 633 nm) was used for analyzing, and an Aria III flow cytometer with four lasers (375, 488, 561, and 633 nm) was used for sorting. The analyses were performed using FACSDiVa and the FlowJo (Tree Star) software.

**Cell culture**. MLL-AF9-induced AML cells were maintained in IMDM (Gibco) supplemented with 15% FBS (Gibco), 10 ng/mL mIL-6, 10 ng/mL mIL-3 and 50 ng/mL mSCF (PeproTech). Mouse HSPCs were cultured in IMDM (Gibco) supplemented with 15% FBS (Gibco), 10 ng/mL mIL-6, 10 ng/mL mIL-3, 50 ng/mL SCF, 20 ng/mL thrombopoietin (TPO, PeproTech) and 10 ng/mL Flt3 ligand (Flt3-L, PeproTech).

**Small-molecule compound**. Chaetocin was purchased from Santa Cruz (sc-200893).

**In vivo macrophage depletion**. Macrophages were depleted in leukemia-bearing NOD/SCID mice (sublethally irradiated) using the following treatment procedure: 200 µL of either clodronate or control liposomes was injected intravenously via the tail 2 days before Dox treatment. Then, 100 µL of either clodronate or control liposomes was injected in the same manner on days 3 and 6 after the initiation of the daily Dox treatment. Finally, the mice were sacrificed on day 14 to assess the leukemic burden.

**In vivo NK cell depletion**. NOD/SCID mice received one intraperitoneal injection of 200 µg of purified anti-CD122 antibody 13 days after leukemia cell engraftment. The anti-CD122 monoclonal antibody was generated from the hybridoma cell line TM-β1 (a gift from Dr. Fengchun Yang).

**Human samples**. Normal cord blood mononuclear cells were obtained from the Tianjin Central Hospital of Gynecology and Obstetrics. Primary human AML blasts were obtained from SKLEH's experimental pathology cell bank. The CD34⁺ cells were enriched using a CD34 MicroBead Kit (Miltenyi). For human cell liquid cultures, normal or leukemic CD34⁺ cells were cultured in IMDM with 15% FBS (Gibco), 1% penicillin/streptomycin (Gibco), 100 ng/mL human SCF, 100 ng/mL human Flt3 ligand, 50 ng/mL human TPO, 10 ng/mL human IL-3 and 100 ng/mL human IL-6 (all from PeproTech). For clonogenic assays, normal or leukemic CD34⁺ cells were grown in methylcellulose medium H4435 (StemCell Technologies) with 1% penicillin/streptomycin (Gibco). According to the regulations of the institutional ethics review boards from the Institute of Hematology and Blood Diseases Hospital, Chinese Academy of Medical Sciences and Peking Union Medical College, informed consent was signed by all patients

**Electron microscopy**. The sorted GFP⁺ leukemia cells from the control and Dox-treated mice were pelleted for 6 min at 4 °C at 350 xg and fixed in 0.1 M Na cacodylate (pH 7.4) containing 2% glutaraldehyde and 1% PFA at 4 °C for 1 day. Then, the samples were submitted to the Electron Microscopy Core Facility of the SKLEH for standard transmission electron microscopy ultrastructural analyses.

**Mouse CFC assay**. GFP⁺ leukemia cells were sorted and cultured in methylcellulose-based medium (3231, StemCell Technologies), which included 10 ng/mL mIL-6, 10 ng/mL mIL-3, 50 ng/mL mSCF and 10 ng/mL granulocyte-

macrophage colony-stimulating factor (GM-CSF). Then, the cells were plated in 24-well plates in 0.5 mL of media at a density of 400 cells/mL with 4–6 replicate wells. CFCs were scored under an inverted microscope after 7–10 days of incubation. Normal HSPCs were enriched and cultured in methylcellulose-based medium (3434, StemCell Technologies) in 24-well plates at appropriate numbers per well. The colonies were counted and collected after 7–14 days of incubation. Colony types A, B, and C were characterized according to the morphology: Type A colonies are very compact without a halo of migrating cells; Type B colonies have a compact center and a halo of single cells; Type C colonies have no center and only single cells[54].

**Generation of iPSCs**. Mouse iPSCs were maintained in standard mouse ES cell culture medium. Primary mouse embryonic fibroblasts (MEFs) were obtained from 13.5-day embryos of ICR mice based on the protocol from Wicell (Madison, WI) and cultured in DMEM containing 10% FBS. Mouse iPSCs were cultured on the mitomycin C-treated MEFs (10 µg/ml). All types of cells were cultured in 5% $CO_2$ at 37 °C in ES culture medium. The GFP⁺DAPI⁻ leukemia cells were cultured with 2 µg/mL Dox, 50 ng/mL mSCF, 10 ng/mL mIL-3 and 10 ng/mL mIL-6. Mouse HSPCs were cultured with 2 µg/mL Dox, 10 ng/ml mIL-6, 10 ng/mL mIL-3, 50 ng/mL mSCF, 20 ng/mL mTPO and 10 ng/mL mFlt3-L. Granulocytes were cultured with 2 µg/mL Dox, 10 ng/mL mG-CSF and 5 ng/mL mGM-CSF. Cytokines were removed from the culture system after 7 days, and the cells were cultured only in the presence of Dox in ES culture medium. Dox was withdrawn after ES-like colonies came up.

**Western blot**. Cell extracts were prepared using RIPA buffer, resolved on NuPAGE 6–12.5% gradient Bis-Tris gels, transferred to nitrocellulose and hybridized using antibodies against p53 (Cell Signaling, 2527S, 1:500), Puma (Abcam, ab9643, 1:500), caspase-3 (Cell Signaling, 9662S, 1:1,000), β-actin (Cell Signaling, 3700S, 1:5,000), H3 (Abcam, ab1791, 1:5,000), H3K9me3 (Abcam, ab8898, 1:1,000), Sox2 (Abcam, ab93689, 1:1,000), H3K4me3 (Abcam, ab8580, 1:1,000), H3K27me3 (Abcam, ab6002, 1:1,000), H3K79me2 (Abcam, ab3594, 1:1,000) and H3K36me3 (Abcam, ab194677, 1:1,000). Uncropped scans of the most important blots are displayed in Supplementary Fig. 10.

**Microarray**. Microarray experiments were conducted using nine in vivo samples, including three Dox-untreated leukemia cell controls and three 48 h and 72 h Dox-induced leukemia cell samples. Microarray experiments were performed by CnKingBio in Beijing. Total RNA was isolated with TRIzol reagent (Invitrogen, Canada) and purified using an RNeasy Mini Kit (Qiagen, German); the purification included a DNase digestion treatment step. RNA concentrations were determined using a NanoDrop 2000 spectrophotometer (Thermo, USA). RNA was labeled using a GeneChip® WT Terminal Labeling and Controls Kit combined with an Ambion® WT Expression Kit. Labeled DNA was hybridized to Mouse Gene 2.0 ST GeneChip® arrays (Affymetrix, USA). Oligo package[55] was used to extract CEL files and normalize the microarray signal by RMA method. Limma package[56] was used to identify significantly differentially expressed genes form 0 h, B-H method was used for p-value correction with an FDR of 0.05 as statistically significant. Finally, differential expressed genes were used clusterProfiler package[57] to do biological process enrichment.

**RNA-sequencing**. Both MLL-AF9-OSKM and tetO-OSKM cKit⁺ cells that underwent 0, 3, 6, 12 and 24 h of Dox treatment (2 µg/mL) were retrieved and followed with RNA-seq library construction before high-throughput sequencing. The experiment has two biological replicate. RNA samples were prepared using rRNA depletion and strand-specific library construction method. RNA was extracted using TRIzol™ (Invitrogen) and followed by rRNA removal (Ribo-Zero™ rRNA Removal Kits) before library construction (NEBNext® Ultra™ Directional RNA Library Prep Kit for Illumina® Kit). Illumina TrueSeq v2 protocol was used on HiSeq XTen with paired-end read of 150 bp + 8 bp index. Sequencing were performed by NovoGene company in Beijing. Reads were aligned to mm10 mouse genome assembly by STAR package[58] using ENCODE recommended parameters. Samtools package[59] was used to select unique and proper paired reads, and the count table was generated using HTSeq package[60]. Counts were normalized, and differential expression analysis was performed by DEseq2 package according to Love et al.[61]. B-H method was used for p-value correction with an FDR of 0.05 as statistically significant. To reveal the true effect of OSKM induction, a confidence interval was used to remove the unwanted noise inherent in cell culture. Briefly, for each paired gene in the same time point, the gene expression fold change (compared to 0 h) ratio between Dox group and control group was calculated and named by r. Next, $r_{rep1}$ and $r_{rep2}$ were used to calculate the confidence interval and genes whose confidence interval region didn't contain 1 were considered as OSKM affected differentially expressed genes.

The OSKM direct-effect genes were assigned to whichever differentially expressed genes whose promoter region contain either Oct4, Sox2, Klf4, c-Myc or Oct4-Sox2 motifs by using the Homer package[62]. The biological processes for these OSKM direct-effect genes were searched using the metascape package according to Tripathi et al.[63] before performing cluster analysis using ComplexHeatmap[64].

**Chromatin openness analysis**. 50,000 cells were lysed in lysis buffer (10 mM Tirs-HCl, PH 7.4, 10 mM NaCl, 3 mM MgCl$_2$, 0.1%(v/v) IGPAL CA-630) for 10 min on ice before centrifuged at 500 xg for 5 min. The nuclei were added with 50 μL transposition reaction buffer (5 μL TruePrep Tagment Enzyme,10 μL TruePrep Tagment Buffer L and 35 μL ddH$_2$O from Vazyme TD501-01) and followed with incubation at 37ºC for 30 min. After tagmentation, VAHTS DNA Clean Beads were used to stop the reaction and DNA was purified for final library construction (TruePrep$^{TM}$ DNA Library Prep Kit V2 for Illumina) before paired-end high-throughput sequencing using HiSeq XTen.

Reads were aligned to mm10 mouse genome assembly by BWA package[59] and followed with unique and proper paired reads selection using the Samtools package[59]. Peaks were called using macs2 package[65]. The DiffBind package[66] was used to find CO peaks (peaks at later time point have higher occupancy than zero time point) or OC (peaks at later time point have smaller occupancy than zero time point) with the false discovery rate > 5%. The expression level of the closest TSS, within a 2 kb window around each peak, was considered to be the regulated gene for that particular peak. The biological processes for these genes were found using the metascape package according to Tripathi et al.[63] before performing cluster analysis using ComplexHeatmap[64].

In order to determine which transcription factors may play an important role in differential peak regions, the Homer package[62] was used to calculate the significance (p-value) for all known transcription factor motifs at those peaks for all time points. We assumed that if a transcription factor played an important role, its motif would have the most fluctuated p-value. The standard deviation of p-value from each motif among all time points was calculated, and the top 20 motifs with the largest variance were plotted in Fig. 5a.

**RNA extraction and qRT-PCR**. Total RNA was isolated using a Qiagen RNeasy Mini Kit. cDNA was synthesized using Improm-IITM reverse transcriptase (Promega). qRT-PCR was performed on a 7500 or StepOne real-time PCR system (Applied Biosystems). Primers used in this study were: mouse *Gapdh*: TGGC AAAGTGGAGATTGTTGCC and AAGATGGTGATGGGCTTCCCG; mouse *Oct4*: ACATCGCCAATCAGCTTGG and AGAACCATACTCGAACCACATCC; mouse *Sox2*: ACAGATGCAACCGATGCACC and TGGAGTTGTACTGCAG GGCG; mouse *Klf4*: GCACACCTGCGAACTCACAC and CCGTCCCAGTCA CAGTGGTAA; mouse *c-Myc*: ACCACCAGCAGCGACTCTGA and TGCCTCTT CTCCACAGACACC; mouse *Kdm3a*: TTGGCAGAACGAAAATCACCT and CCGAACAGAAGTTATGGCTCC; human *Gapdh*: GAAGGTGAAGGTCGGAG TC and GAAGATGGTGATGGGATTTC; human *Oct4*: GCTCGAGAAGGATGT GGTCC and CGTTGTGCATAGTCGCTGCT; human *Sox2*: CACTGCCCCTCTC ACACATG and TCCCATTTCCCTCGTTTTTCT; human *Klf4*: AGAGTTCCCA TCTCAAGGCA and GTCAGTTCATCTGAGCGGG; human *c-Myc*: TTTCGGG TAGTGGAAAACCA and CACCGAGTCGTAGTCGAGGT

**Statistical analysis**. GraphPad Prism software was used for the statistical analyses. Student's t test was used for two-group comparisons, and ANOVA with a Bonferroni post-hoc correction was used for three-group comparisons. A p-value < 0.05 was considered significant.

**Reporting summary**. Further information on research design is available in the Nature Research Reporting Summary linked to this article.

## Data availability

The microarray, RNA-seq, and ATAC-seq raw data that support the findings of this study have been deposited in the Gene Expression Omnibus database (www.ncbi.nlm.nih.gov/geo) under accession numbers GSE64414 and GSE121123.

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

## Acknowledgements

The authors are grateful to our lab members for their assistance with the experiments. We appreciate insightful suggestions from Drs. Dangsheng Li, Emery Bresnick, David Scadden, Hongkui Deng, Jian Yu, Lin Zhang and Shiyuan Cheng, and proof reading by Ms. Amy Cheng. This work was supported by grants from the Ministry of Science and Technology of China (2016YFA0100600, 2017YFA0103400), the National Natural Science Foundation of China (81421002, 81430004, 81922002, 81890990, 81730006, 81861148029, 81870086, 81700164, 81700166, 31522031, 31571526, 81900113). CAMS Initiative for Innovative Medicine (2017-I2M-3-009, 2016-I2M-1-017), CAMS Fundamental Research Funds for Central Research Institutes (2018PT31005, 2017PT31032), SKLEH-Pilot Research Grant (Zk18-02, Zk18-05, Zk17-04) and China Postdoctoral Science Foundation 2019M650573.

## Author contributions

YJ.W., T.L., GH.S., and YW.Z. designed and performed the experiments, analyzed the data and wrote the paper. SD.Y., HY.Z., S.H., and YF.L. helped with the in vivo mouse experiments and iPS work. SH.M. performed the two-photon imaging. HY.Z. helped with bioinformatics analysis. YX.R. performed the electron microscopy. SR.G. provided the OSKM mice and assisted with the manuscript. KY.Y., H.C., and T.C. proposed the study, designed the experiments, interpreted the results, wrote the paper, and oversaw the research project.

## Competing interests

The authors declare no competing interests.
