## [Peer Review File · Nature Communications]

Reviewers' comments:

Reviewer #1 (Remarks to the Author):

Review of NCOMMS-19-01447 – Targeting of apoptosis gene loci by reprogramming factors leads to selective eradication of leukemia cells

In this manuscript, Cheng et al. claim that that expression of transcription factors used for induced pluripotent stem (iPS) cell reprogramming (Oct4, Sox2, Klf4 and Myc, termed OSKM) selectively induce apoptosis of Acute Myeloid Leukemia (AML) cells in vivo and in vitro while sparing normal hematopoietic cells. Using bone marrow transplantation models of MLL-fusion-induced AML in a mouse strain of Doxycycline (Dox)-inducible OSKM expression, they show that induction of OSKM factors caused rapid depletion of GFP-positive MLL-AF9-expressing leukemia cells in non-competitive as well as competitive settings. As the effects were preserved in immuno-compromised mouse strains and/or upon depletion of immune cell subsets, the authors rule out a mechanistic involvement of the immune system in this phenomenon. Time-resolved gene expression profiling after OSKM induction shows that more extensive transcriptional changes are observed in MLL-AF9-AML cells vs. wild-type cells over a time course of 24 hrs. OSKM induction in MLL-AF9- but not in wild-type cKit+ progenitors caused rapid induction of p53, cleavage of caspase 3 and apoptosis. ATAC-seq analysis over the same time course shows that MLL-AF9 cells re-organize global genome accessibility more extensively than wild type cKit+ cells, which includes selective opening of loci that correspond to apoptosis regulators. Based on OSKM-induced modulation of global H3K9me3 levels that are found in MLL-AF9 but not wild-type cells, the authors speculate over an involvement of Suv39h1/Suv39h2 enzymes in this mechanism. Finally, they show that Sox2 and Klf4 are sufficient to induce specific killing of MLL-AF9 cells, while Oct4 and Myc expression alone did not have an effect.

While this manuscript contains a lot of dataset and analyses that describe an interesting phenomenon, the work suffers from several major drawbacks. First: the research question behind this work is not entirely clear. While the authors have shown in earlier work using the same model that MLL-AF9-cells can be reprogrammed to iPS cells (Liu et al, Leukemia 2014), they now state that OSKM expression induces apoptosis in MLL-AF9-transduced cells. Second: While global RNA-seq and ATAC-seq analyses identify extensive re-shaping of transcriptional programs and genome accessibility that appears to be specific for MLL-AF9-AML cells, the work does not provide mechanistic insight into why the observed effect is specific to leukemia cells and does not affect normal cells.

The authors state several times throughout the manuscript that based on their findings, reprogramming mechanisms could be envisaged for cancer therapeutics. However, given the previous results published by the same group (Liu et al, Leukemia 2014) and the lack of a clear mechanistic understanding of the observed effects, this is not justified.

Major points:

- The majority of the work uses a transplantation model that is based on retroviral MLL-AF9 expression. Only few results are provided from other mouse models. More data from other mouse leukemia models (using different oncogenic drivers) as well as from human (normal and cancer) cells are required to establish the general relevance of the proposed concept.
- Time course experiments in Figures 4B-E and 6B-C are only performed for up to 4 days. As wild-type cKit+ cells appear to undergo apoptosis and cease to proliferate upon OSKM /Sox2+Klf4 expression at later time points, the question arises whether the response is just delayed in wild type cells. A longer time course is required to clarify this.

- The transient decrease in H3K9me3 / Suv39h1/Suv39h2 expression in MLL-AF9 but not wild type cells is interesting. Yet, the mechanistic implications of this effect are not followed up. Why does the change in Suv39h1/Suv39h2 RNA is shown only at 24 hrs after OSKM induction if the WB shows a decrease in H3K9me3 already at 6/12 hrs?

Does overexpression of Suv39h1/Suv39h2 rescue the OSKM sensitivity in MLL-AF9 cells, or vice versa, does Suv39h1/Suv39h2 knockdown in wild type cells sensitize them to OSKM-induced apoptosis?

Is the same effect on gene expression/chromatin accessibility/H3K9me3/ Suv39h1/Suv39h2 expression observed in MLL-AF9 cells upon Sox2+Klf4 expression?

- The authors report massive transcriptional and epigenomic reorganization during OSKM induction in MLL-AF9 vs. wild type cells. In that context it would be interesting to show the differences in transcriptomes and genome accessibility between MLL-AF9- and wild type cells before OSKM induction (i.e. differences in the steady state).

- There is no intersection of ATAC-seq and RNA-seq data. It would be very interesting to show how changes in chromatin accessibility correlate with gene expression changes, both on a global scale as well as on selected examples of genes. For instance, the authors could include information on RNA-expression in the examples shown in Supplementary Figure 5B and C.

Minor points:

In addition to some grammatical and stylistic errors in the text, the manuscript suffers from several inconsistencies and mistakes that need to be addressed (e.g. the use of ckit+ vs. cKit+ throughout the manuscript).

- Figure 1d: no yellow arrows are visible, despite they are mentioned in the figure legend. The scale bar is too small to see.

- Figure 2: Which ratios of MLL-AF9 vs. wild type cells were co-transplanted? What was the percentage of tetO-OSKM wild type cells when the MLL-AF9-positive population was at 20% / 50%? Only the fold change is shown in figure 2C.

- Figure 2E: axis label of right panels: tetO-OSKM: a „K“ is missing

- The legend to Figure 2B refers to orange traces, but no orange traces are shown in the figure.

- Figure 3A and legend: What is the gray/black distinction in the annotation of biological pathways supposed to indicate?

- Figure 4A and legend: It is not clear which subset of genes were used for this analysis. The legend only states: “differentially expressed genes that contain OSKM motifs inside promoter regions”. Were differentially expressed genes called for MLL-AF9 cells only (and their regulation plotted for cKit+ cells), or was differential gene expression analysis performed for cKit+ cells in parallel? Please clarify.

- Figure 4D: It is not clear how C3AI-positive colonies were picked from the CFC assay? According to the Methods section, the C3AI indicator plasmid contains a Puromycin resistance gene.

- Figure S4B: Was batch correction used to analyse these datasets? If yes, which algorithm?

- Legend to Figure 5A: The description of left and right panels is switched.

- Figure 5B: A different color scale should be used for cKit+ cells. In the text the authors claim that the Gene Ontology Term “cell death” first goes up and then down, but this cannot be seen in the figure.

- Fig 5D: A “-” sign is missing in the right panel for the cKit+ cells. In addition, it is not entirely clear that “-” and “+” signs refer to Dox treatment in this figure. Please clarify by improving the annotation of the figure and/or in the figure legend.

- Figure S5A: What does “rows not linked” in the heading of the figure mean?

- The data in Figures S5B-C are not explained for MLL-AF9 cells in the manuscript text.

Reviewer #2 (Remarks to the Author):

This is an improved version of an excellent manuscript that i reviewed in the past for a different journal. the authors systematically study the impact of inducing OSKM in normal and leukemic cells. The authors establish that OSKM eradicates leukaemia cells by depleting H3K9me3 and subsequent induction of apoptosis promoting genes. They now show that Sox2 and Klf4 predominantly underlie the observed effect.

The manuscript is clearly written and all conclusions are novel and well supported by the data presented. The techniques and tools are cutting edge, and references are relevant and balanced. The authors have addressed a point i raised in the past about delineating which of the yamanaka factors contribute to the observed phenotype.

I have no further comments or requests to improve this solid and interesting manuscript.

Point-by-point Response to Reviewers

Reviewer #1

In this manuscript, Cheng et al. claim that that expression of transcription factors used for induced pluripotent stem (iPS) cell reprogramming (Oct4, Sox2, Klf4 and Myc, termed OSKM) selectively induce apoptosis of Acute Myeloid Leukemia (AML) cells in vivo and in vitro while sparing normal hematopoietic cells. Using bone marrow transplantation models of MLL-fusion-induced AML in a mouse strain of Doxycycline (Dox)-inducible OSKM expression, they show that induction of OSKM factors caused rapid depletion of GFP-positive MLL-AF9-expressing leukemia cells in non-competitive as well as competitive settings. As the effects were preserved in immuno-compromised mouse strains and/or upon depletion of immune cell subsets, the authors rule out a mechanistic involvement of the immune system in this phenomenon. Time-resolved gene expression profiling after OSKM induction shows that more extensive transcriptional changes are observed in MLL-AF9-AML cells vs. wild-type cells over a time course of 24 hrs. OSKM induction in MLL-AF9- but not in wild-type cKit+ progenitors caused rapid induction of p53, cleavage of caspase 3 and apoptosis. ATAC-seq analysis over the same time course shows that MLL-AF9 cells re-organize global genome accessibility more extensively than wild type cKit+ cells, which includes selective opening of loci that correspond to apoptosis regulators. Based on OSKM-induced modulation of global H3K9me3 levels that are found in MLL-AF9 but not wild-type cells, the authors speculate over an involvement of Suv39h1/Suv39h2 enzymes in this mechanism. Finally, they show that Sox2 and Klf4 are sufficient to induce specific killing of MLL-AF9 cells, while Oct4 and Myc expression alone did not have an effect.

While this manuscript contains a lot of dataset and analyses that describe an interesting phenomenon, the work suffers from several major drawbacks. First: the research question behind this work is not entirely clear. While the authors have shown in earlier work using the same model that MLL-AF9-cells can be reprogrammed to iPS cells (Liu et al, Leukemia 2014), they now state that OSKM expression induces apoptosis in MLL-AF9-transduced cells. Second: While global RNA-seq and ATAC-seq analyses identify extensive re-shaping of transcriptional programs and genome accessibility that appears to be specific for MLL-AF9-AML cells, the work does not provide mechanistic insight into why the observed effect is specific to leukemia cells and does not affect normal cells.

The authors state several times throughout the manuscript that based on their findings, reprogramming mechanisms could be envisaged for cancer therapeutics. However, given the previous results published by the same group (Liu et al, Leukemia 2014) and the lack of a clear mechanistic understanding of the observed effects, this is not justified.

***Response:** We apologize for the confusion. In our previous work (Liu et al., Leukemia 2014), MLL-AF9 cells were used to study whether leukemia cells can be reprogrammed into cancer-iPSC in vitro. Albeit successful, the reprogramming can only be achieved when grown under embryonic stem cell medium culture. If the reprogramming process*

was initiated in hematopoietic system, we observed a complete elimination of MLL-AF9 cells. Around that time, Serrano group reported that transient OSKM treatment in vivo reprogrammed normal somatic cells into iPSCs (Abad et al., 2013) whereas Yamada group reported that transient expression of OSKM in vivo leads to cancer development in various tissues (Ohnishi et al., 2014). These phenomena prompted our research question as to what happens to the established or already transformed cancer cells (not the normal or pre-malignant cells) if treated with OSKM in vivo.

Strikingly, instead of reprogramming, in vivo transient OSKM treatment caused MLL-AF9 cell number reduction and prolonged leukemia mice survival. During in vivo OSKM induction, we observed increase in chromatin accessibility at genes encoding apoptotic regulators, decreased H3K9me3 level and upregulation of histone demethylase KDM3A. This OSKM-induced apoptosis phenotype can be mimicked when treated MLL-AF9 AML cells with chaetocin, a small-molecule inhibitor of H3K9 methylation. Furthermore, this OSKM sensitivity of leukemia cells can be partially rescued by inhibition of histone demethylase KDM3A. Our results suggest a possible mechanism where transient in vivo OSKM treatment induced H3K9me dysregulation that cause selective killing in MLL-AF9 AML cells but not in normal cKit⁺ HSPC cells.

Furthermore, our work showed that transient OSKM induction can selectively activate apoptosis in leukemia cells when grown in their natural habitat, either in vivo or in in vitro hematopoietic medium culture. OSKM are known oncogenes, and numerous works speculate on the relation between reprogramming process and cancer development. This work, on the other hand, is the first to demonstrate this selective elimination effect of oncogenes on those established or already transformed cancer cells, which could be envisaged for cancer therapeutics.

Major points:

- The majority of the work uses a transplantation model that is based on retroviral MLL-AF9 expression. Only few results are provided from other mouse models. More data from other mouse leukemia models (using different oncogenic drivers) as well as from human (normal and cancer) cells are required to establish the general relevance of the proposed concept.

Response: *We appreciate the reviewer's comment. In this study, apart from MLL-AF9 induced AML leukemia mice model, Notch1-ALL and NRIP3-AML leukemia mice models and human THP-1 leukemia cell line all displayed OSKM-induced cell number reduction (Fig S1b and new Fig S3 in MS). These phenomena indicate a broader application of this OSKM-induced killing effect, not only in one type of leukemia but several different types and it is cross-species. Nevertheless, it is not our intention to proclaim the observed OSKM-induced apoptosis seen in MLL-AF9 leukemia mice model is the mechanism shared among all tested types of leukemias. Furthermore, it is beyond the scope of this study to examine multiple types of leukemias in parallel. It is for these reasons that we used survival curve, but not delved into the mechanism, to establish the general relevance of this OSKM-induced killing effect in tested types of leukemias.*

- Time course experiments in Figures 4B-E and 6B-C are only performed for up to 4 days. As wild-type cKit⁺ cells appear to undergo apoptosis and cease to proliferate upon OSKM /Sox2+Klf4 expression at later time points, the question arises whether the response is just delayed in wild type cells. A longer time course is required to clarify this.

Response: *We agreed with the reviewer that long-term OSKM induction may affect the function or survival of normal HSPCs. Yet, our purpose is not to examine if the OSKM-induced killing effect may be delayed in normal HSPCs, but rather to find the optimal time window in which leukemia cells can be killed but normal cells still survive.*

- The transient decrease in H3K9me3 / Suv39h1/Suv39h2 expression in MLL-AF9 but not wild type cells is interesting. Yet, the mechanistic implications of this effect are not followed up. Why does the change in Suv39h1/Suv39h2 RNA is shown only at 24 hrs after OSKM induction if the WB shows a decrease in H3K9me3 already at 6/12 hrs?

Does overexpression of Suv39h1/Suv39h2 rescue the OSKM sensitivity in MLL-AF9 cells, or vice versa, does Suv39h1/Suv39h2 knockdown in wild type cells sensitize them to OSKM-induced apoptosis?

Response: *The reviewer raised a critical question about the time difference between transcriptome change in Suv39h1/Suv39h2 and H3K9me3 level change. As Suv39h1/Suv39h2 are not the only contributors that affect H3K9me3 level, we thought to systematically examine the transcriptome change for all H3K9 methyltransferase and demethylases. Besides Suv39h1/Suv39h2, expression of H3K9me3 demethylase KDM3A was progressively up-regulated after 3 hours of OSKM induction (**new Fig 5d** in MS). We first tried to rescue the OSKM sensitivity in MLL-AF9 cells using overexpressed Suv39h1/Suv39h2, but with no success. Furthermore, knockdown of Suv39h1/Suv39h2 in cKit⁺ cells could not sensitize them to OSKM-induced apoptosis. Nevertheless, the OSKM sensitivity in MLL-AF9 cells can be rescued when knockdown KDM3A (**new Fig 5e** and **S7d-e** in MS). In addition, overexpression of KDM3A in normal cKit⁺ HSPCs led to apoptosis and caused cell death effect (**new Fig 5f** in MS). This argued for the involvement of KDM3A, but not Suv39h1/Suv39h2, in OSKM-induced apoptosis. We apologize for the misinterpretation of the data and have corrected our conclusion in the text accordingly.*

*To functionally explore whether H3K9 demethylation selectively affect leukemia cells, we treated AML cells, as well as normal cKit⁺ HSPCs with a small-molecule inhibitor of H3K9 methylation, chaetocin (**new Fig S9a** in MS) (Greiner et al., 2005; Greiner et al., 2013; Lakshmikuttyamma et al., 2010). In the hematopoietic medium culture treated with different concentrations of chaetocin, we found that the number of viable MLL-AF9 cells was significantly reduced by chaetocin, whereas the growth of cKit⁺ cells was only slightly affected, especially at the concentration of 30 nM (**new Fig 7a** in MS). An analysis of apoptosis showed that the chaetocin treatment led to markedly increased apoptosis of MLL-AF9 cells; however, at the same concentration, the apoptotic rate of normal cKit⁺ cells was much lower (**new Fig 7b** in MS). Moreover, in MLL-AF9 cells, 30 nM of chaetocin led to the loss of clonogenic potential (**new Fig 7c** in MS). However,*

the cKit⁺ cells showed only a modest reduction in colony-forming ability (**new Fig 7c** in MS). Therefore, H3K9 methylation activity appears to be essential for the survival of MLL-AF9 cells but less critical for that of normal hematopoietic cells. Importantly, similar results were also obtained in human samples that received this treatment. When human AML CD34⁺ cells and normal CD34⁺ cells were treated with 30 nM of chaetocin, a selective anti-proliferative effect on the primary human AML cells was observed (**new Fig 7d-e** in MS). In addition, the colony-forming ability of the AML cells was more significantly inhibited than was that of the normal CD34⁺ cells (**new Fig S9b-c** in MS). Therefore, H3K9 methylation activity appears to be essential for the survival of AML cells but less critical for that of normal hematopoietic cells.

Is the same effect on gene expression/chromatin accessibility/H3K9me3/Suv39h1/Suv39h2 expression observed in MLL-AF9 cells upon Sox2+Klf4 expression?

Response: To answer this question, we performed RNA-seq/ATAC-seq/Western Blot as described in the Method section on MLL-AF9 cells that harbored either Sox2+Klf4 plasmid or empty vector as a negative control. As shown in Fig 1a below, red font represents similar GO-enriched functions as observed in OSKM. In transcriptome change, functions involved in apoptosis were enriched when induced Sox2+Klf4 alone in MLL-AF9 cells (Fig 1a below). However, we did not observe similar effect in the chromatin accessibility level (Fig 1b below). Since gene expression level does not always correlate with chromatin accessibility level (Huebert et al., 2012; Hendrickson et al., 2018), plus our western blot result showed that H3K9me3 level was decreased (Figure 1c below) and RNA-seq data showed the expression of KDM3A was also increased (Figure 1d below), we hence concluded that the induction of Sox2+Klf4 alone causes similar effects as OSKM do in MLL-AF9 cells.

Figure 1. The enriched biological functions in overexpressed Sox2 and Klf4 in MLL-AF9 cells

a) Gene Ontology (GO) analysis on differentially expressed genes that contain SK motifs inside promoter regions. Heatmap of the top 20 enriched biological pathways. Red font represents similar GO-enriched functions as in Fig 4a in MS.

b) Gene Ontology (GO) analysis on chromatin differential openness regions. Heatmap of the top 20 enriched biological pathways. Red font represents similar GO-enriched functions as in Fig 5b in MS.

c) Western blot of H3K9me3 in AML cells transduced with S+K. The data shows the results after Dox induction at different time points.

d) KDM3A expression after S+K transduction.

- The authors report massive transcriptional and epigenomic reorganization during OSKM induction in MLL-AF9 vs. wild type cells. In that context it would be interesting to show the differences in transcriptomes and genome accessibility between MLL-AF9- and wild type cells before OSKM induction (i.e. differences in the steady state).

Response: To assess the spectrum of the differences, we focused on two different perspectives: one is the gene level that translates into biological functions, and the other is at transcription regulation level where we focus on the role of transcription factors (TFs).

Using differentially behaved genes to enrich for significant biological functions, we found

that 1) half of the enriched biological functions from both transcriptome and chromatin openness assay are overlapped (as shown in red in Fig 2 below), and 2) more than half of the enriched biological functions are related to immune response and leukocyte. As one of the significant phenotypes in leukemia is the aberrant leukocyte and its associated immune response, our function enrichment results are consistent with the current knowledge.

We then moved to the transcription regulation level. Under the assumption that nearly all TFs have their preferred DNA binding motifs and a majority of them bind to cis-regulatory regions that can be identified via ATAC-seq, we characterized which TFs are enriched in cis-regulatory regions using motif analysis. Using chromVAR package as described in Schep et al., 2017, we first retrieved top 20 TFs whose DNA binding motif occurrence in aggregated chromatin accessible regions was different between AML and cKit⁺ cells. Depending on its expression level, only 10 transcription factors showed a significantly different expression level between these two cell types. Among them, Hoxa9 was the most differentially behaved transcription factor. Its role in MLL-rearranged leukemia, especially in AML cells, are well documented (Bernt et al., 2011; Collins and Hess., 2016). Hematopoietic-specific transcription factors, such as Gata1, Klf1 and Bcl6, were mainly enriched in cKit⁺ cells (Table 1 below). All these suggest that, at the steady/ground state, AML and cKit⁺ cells have their preferential transcription factors.

Figure 2. The enriched biological functions in MLL-AF9 AML and cKit⁺ cells at the steady-state. The top panel was the biological function enrichment result using differentially expressed genes calculated via transcriptome analysis, and the lower panel used differential accessible regions via chromatin openness analysis.

Table 1. The differential transcription factors between MLL-AF9 AML and cKit⁺ cells

JASPAR ID	TF Name	Function	MLL-AF9	cKit ⁺
MA0594.1	Hoxa9	leukemia/HSC	High	Low
MA0039.2	Klf4		High	Low
MA0147.2	Myc/c-Myc		High	Low
MA0519.1	Stat5a::Stat5b	proliferation/apoptosis	Low	High
MA0493.1	Klf1	hematopoietic development	Low	High
MA0742.1	Klf12	development and Tumour-suppression	Low	High
MA0035.3	Gata1	development/erythroblasts	Low	High
MA0150.2	Nfe2l2	stress response	Low	High
MA0463.1	Bcl6	B cell	Low	High
MA0518.1	Stat4	T cell	Low	High
MA0816.1	Ascl2	reinforces intestinal stem cell identity	q > 0.05	q > 0.05
MA0099.2	FOS::JUN	--		
MA0591.1	Bach1::Mafk	--/hematopoietic		
MA0485.1	Hoxc9	leukemia/HSC	0	0
MA0910.1	Hoxd8	leukemia/HSC	0	0
MA0913.1	Hoxd9	leukemia/HSC	0	0
MA0482.1	Gata4	development	0	0
MA0047.2	Foxa2	development	0	0
MA0499.1	Myod1	muscle development	0	0
MA0500.1	Myog	muscle differentiation	0	0

- There is no intersection of ATAC-seq and RNA-seq data. It would be very interesting to show how changes in chromatin accessibility correlate with gene expression changes, both on a global scale as well as on selected examples of genes. For instance, the authors could include information on RNA-expression in the examples shown in Supplementary Figure 5B and C.

Response: To address how changes in chromatin accessibility correlate with gene expression changes, we thought to examine this correlation using direct observation and quantification method.

We employed an aggregated gene list that contains all differentially expressed genes, regardless of which timepoints after OSKM induction, in MLL-AF9 AML and cKit⁺ cells separately. Differentially expressed genes were calculated as mentioned in Method. We then pull out each gene-associated chromatin accessibility at different time points after OSKM induction. To avoid possible skew from existing outliers, we employed standard score (z-score) to measure transcriptome and chromatin accessibility change for each differentially expressed gene. We then plotted these z-scores as a heatmap in the left panel of Fig 3a-b below for intuitive observation. To give a more quantitative measure, we examined all pair-wise transcriptome-chromatin-accessibility comparison of these differentially expressed genes using z-score. The right panel of Fig 3a/b below is the heatmap representing Pearson correlation R^2 values for correlation levels.

We observed both positive and negative correlation for those differentially expressed genes. There are more positively correlated genes in MLL-AF9 AML cells than cKit⁺ cells. Regardless of positive or negative correlation, MLL-AF9 AML cells had higher correlation

R^2 values than $cKit^+$ cells. This suggests that the change in MLL-AF9 AML cells is more homogenous than $cKit^+$ cells. Furthermore, in MLL-AF9 AML cells, the correlation between transcriptome and chromatin accessibility can be split into two patterns: one contains 0h, 3h and 6h after OSKM induction; the other one has 12h and 24h (the right panel of Fig 3a-b below). Interestingly, just by looking at transcriptome (**Fig S4a** in MS) or chromatin accessibility alone, 0h and 3h are closer to each other whereas 6h, 12h and 24h are closer within themselves. It is worth noting that, among the 0h and 3h group, regardless of transcriptome or chromatin accessibility, this group is correlated with no OSKM induction control samples. All these make us speculate that the time-window between 6h to 12h after OSKM induction is the crucial moment for MLL-AF9 cell choosing its future fate.

Among the selected examples of genes listed in the manuscript, both in MLL-AF9 and $cKit^+$ cells, *Klf4*, *Myb*, *Oct4*, *Pu.1* and *Sox2* showed a positive correlation between gene expression and chromatin accessibility changes. *Nanog*, on the other hand, only showed chromatin accessibility change without transcriptome change. In Fig 4 below, only *Myb*, *Oct4* and *Nanog* were presented as examples to illustrate the changes.

Figure 3. Correlation of changes in gene expression and its associated chromatin accessibility between MLL-AF9 AML and c-Kit⁺ cells.

a) The left panel is the heatmap of the z-score that measures transcriptome and chromatin accessibility change in MLL-AF9 AML cells whereas the right panel is the heatmap representing Pearson correlation R^2 values for correlation between changes of transcriptome and chromatin accessibility. b) the same as in c-Kit⁺ cells.

Figure 4. Genomic loci and expression view of selected example genes as seen in Fig S6 in MS.

Upper panel is the expression change of Myb, Oct4 and Nanog during different time points after OSKM induction. Lower panel is the same as Fig S5; it presented the IGV view of Myb, Oct4 and Nanog loci. The grey box marked where the changes locate.

Minor points:

In addition to some grammatical and stylistic errors in the text, the manuscript suffers from several inconsistencies and mistakes that need to be addressed (e.g. the use of c-Kit+ vs. cKit+ throughout the manuscript).

Response: We appreciate the reviewer's criticism and we have modified accordingly.

- Figure 1d: no yellow arrows are visible, despite they are mentioned in the figure legend. The scale bar is too small to see.

Response: We appreciate the reviewer's criticism and we have modified the figure according to the reviewer's suggestion.

- Figure 2: Which ratios of MLL-AF9 vs. wild type cells were co-transplanted? What was the percentage of tetO-OSKM wild type cells when the MLL-AF9-positive population was at 20% / 50%? Only the fold change is shown in figure 2C.

Response: 2×10^5 MLL-AF9-OSKM cells and 2×10^7 tetO-OSKM CD45.2⁺ cells were co-transplanted into lethally irradiated (9.5 Gy) B6.SJL CD45.1 recipient mice. We have

modified the figure legend accordingly. When AML cells reached 20% / 50%, the percentage of tetO-OSKM CD45.2⁺ were 76~79% / 46~48%, respectively. As the recipient mice were lethally irradiated, very few recipient CD45.1⁺ cells could be detected.

- Figure 2E: axis label of right panels: tetO-OSKM: a „K“ is missing

Response: *We appreciate the reviewer's criticism and we have modified the figure accordingly.*

- The legend to Figure 2B refers to orange traces, but no orange traces are shown in the figure.

Response: *We appreciate the reviewer's criticism and we have modified the figure legend accordingly.*

- Figure 3A and legend: What is the gray/black distinction in the annotation of biological pathways supposed to indicate?

Response: *We apologize for the confusion. Black font indicates the biological pathways that are related to immune response whereas grey font indicates other biological pathways. We have modified the figure legend accordingly.*

- Figure 4A and legend: It is not clear which subset of genes were used for this analysis. The legend only states: "differentially expressed genes that contain OSKM motifs inside promoter regions". Were differentially expressed genes called for MLL-AF9 cells only (and their regulation plotted for c-Kit⁺ cells), or was differential gene expression analysis performed for c-Kit⁺ cells in parallel? Please clarify.

Response: *We apologize for the confusion. This was the differential gene expression analysis performed for MLL-AF9 and cKit⁺ cells in parallel. The analysis was detailed in Materials. We have modified the figure legend accordingly.*

- Figure 4D: It is not clear how C3AI-positive colonies were picked from the CFC assay? According to the Methods section, the C3AI indicator plasmid contains a Puromycin resistance gene.

Response: *We added puromycin in the CFC culture system, and picked more than 10 single colonies after 10 days culture, before expanded each colony in liquid culture medium with puromycin. We have modified the sentences accordingly.*

- Figure S4B: Was batch correction used to analyse these datasets? If yes, which algorithm?

Response: *We did not see any batch effect in our dataset, so batch correction was not used in our analysis.*

- Legend to Figure 5A: The description of left and right panels is switched.

Response: *We apologize for the confusion. We have corrected the figure legend accordingly.*

- Figure 5B: A different color scale should be used for c-Kit+ cells. In the text the authors claim that the Gene Ontology Term “cell death” first goes up and then down, but this cannot be seen in the figure.

Response: *We appreciate the reviewer’s suggestion. We have already tried different color scales for cKit⁺ cells. However, the enrichment value for “cell death” is still not significant to be seen on the plot. Furthermore, the observation that “cell death” first goes up and then down was shown in transcriptome analysis (Fig 4a). In chromatin accessibility assays, we did not observe “cell death” function in the biological function enrichment assay.*

- Fig 5D: A “-“ sign is missing in the right panel for the c-Kit+ cells. In addition, it is not entirely clear that “-“ and “+“ signs refer to Dox treatment in this figure. Please clarify by improving the annotation of the figure and/or in the figure legend.

Response: *We apologize for the mistakes. We have changed the figure.*

- Figure S5A: What does “rows not linked” in the heading of the figure mean?

Response: *In Fig S5A, each row in the heatmap represents one chromatin open region (peak). The plot was descendingly sorted according to peak value. Therefore, the same row may not contain the same chromatin open region among these different time points. We have modified the figure legend.*

- The data in Figures S5B-C are not explained for MLL-AF9 cells in the manuscript text.

Response: *We apologize for the confusion. We have modified the figure legend.*

Reviewer #2

This is an improved version of an excellent manuscript that i reviewed in the past for a different journal. the authors systematically study the impact of inducing OSKM in normal and leukemic cells. The authors establish that OSKM eradicates leukaemia cells by depleting H3K9me3 and subsequent induction of apoptosis promoting genes. They now show that Sox2 and Klf4 predominantly underlie the observed effect.

The manuscript is clearly written and all conclusions are novel and well supported by the data presented. The techniques and tools are cutting edge, and references are relevant and balanced. The authors have addressed a point i raised in the past about delineating which of the yamanaka factors contribute to the observed phenotype.

I have no further comments or requests to improve this solid and interesting manuscript.

***Response:** Many thanks for the positive assessment about our manuscript from this reviewer.*

REVIEWERS' COMMENTS:

Reviewer #1 (Remarks to the Author):

The authors made a significant effort to address the points I raised in the first round of the revision. The revised version of this manuscript contains several novel datasets and analysis, which contribute to the improvement of the work. I have a few additional suggestions:

- The revised version contains results about the role of the histone demethylase KDM3A in the OSKM-induced killing of AML cells, which are very interesting. Yet, it is not clear why the authors removed the dataset on Suv39h1/Suv39h2 from the manuscript. Even if these results show that these enzymes are not responsible for the effect, it would be of interest to show these results in a supplementary figure.
- I recommend the authors include the results shown in Figure 1c in the point-by-point response to the reviewers in the manuscript in Figure 6 or S8, as it strengthens the evidence for the importance of KDM3A in the process of leukemia eradication by Sox2/Klf4.
- Can the authors show data that prove lentiviral KDM3A over-expression for experiments in Figure 5?
- Can the authors show data that prove lentiviral O/S/K/M over-expression for experiments in Figure 6?
- As the specificity of Chaetocin has been debated, it would help if the authors could include data on the effect of Chaetocin on other histone marks in Figure S9.
- I suggest the authors further tone down the claims about the potential of OSKM-mediated therapy approaches throughout the manuscript. Given the limited evidence of the specificity of the effect, this is not justified.

Minor points:

- Figure 4a - legend: Explain black vs. grey color of GO annotations.
- Figure 5f: FACS data panels are switched, Ctrl should be KDM3A and vice versa.
- Figure 7c and Figure S8: Please explain in the legend what colonies of type A/B/C refers to.
- Figure 7d: Use the same section on the y-axis in left and right panels.
- Figure 7e: A blue dot is missing in the legend for "ctrl".
- Figure S5: Write "without" instead of "w/o".
- Figure S10: As this analysis is done on a cohort of Follicular Lymphoma, it does not add convincing evidence for a role in KDM3A in leukemia. I therefore suggest to remove it.
- Line 237/238: Refer to Figure S5d instead of S4d

Point-by-point Response to Reviewers

Reviewer #1 (Remarks to the Author):

The authors made a significant effort to address the points I raised in the first round of the revision. The revised version of this manuscript contains several novel datasets and analysis, which contribute to the improvement of the work. I have a few additional suggestions:

Response: Many thanks for the positive assessment about our revised manuscript from this reviewer.

- The revised version contains results about the role of the histone demethylase KDM3A in the OSKM-induced killing of AML cells, which are very interesting. Yet, it is not clear why the authors removed the dataset on Suv39h1/Suv39h2 from the manuscript. Even if these results show that these enzymes are not responsible for the effect, it would be of interest to show these results in a supplementary figure.

Response: Kdm3a and Suv39h1/Suv39h2 showed significant dysregulation, but the change in Suv39h1/Suv39h2 occurred much later than the appearance of H3K9 dysregulation (Fig. 6a and Supplementary Fig. 9a-d). We thus focused on Kdm3a, a H3K9 demethylase. Because we think that adding Suv39h1/Suv39h2 data will loosen the logics of this paper, it is not included.

- I recommend the authors include the results shown in Figure 1c in the point-by-point response to the reviewers in the manuscript in Figure 6 or S8, as it strengthens the evidence for the importance of KDM3A in the process of leukemia eradication by Sox2/Klf4.

Response: we agree with the reviewer's suggestion and have added the figure in Supplementary Figure 8b.

- Can the authors show data that prove lentiviral KDM3A over-expression for experiments in Figure 5?

Response: we have added the data in Supplementary Figure 9g.

- Can the authors show data that prove lentiviral O/S/K/M over-expression for experiments in Figure 6?

Response: we have added the data in Supplementary Figure 7a and 7d.

- As the specificity of Chaetocin has been debated, it would help if the authors could include data on the effect of Chaetocin on other histone marks in Figure S9.

Response: we have added the data in Supplementary Figure 8c.

- I suggest the authors further tone down the claims about the potential of OSKM-mediated therapy approaches throughout the manuscript. Given the limited evidence of the specificity of the effect, this is not justified.

Response: we have modified multiple sentences throughout the paper accordingly.

Minor points:

- Figure 4a - legend: Explain black vs. grey color of GO annotations.

Response: we have explained the color in the figure legend.

- Figure 5f: FACS data panels are switched, Ctrl should be KDM3A and vice versa.

Response: we are sorry for the mistake and we have corrected it.

- Figure 7c and Figure S8: Please explain in the legend what colonies of type A/B/C refers to.

Response: we have provided the definition for each type of the colony in the figure legend.

- Figure 7d: Use the same section on the y-axis in left and right panels.

Response: we have changed the y-axis in the right panel.

- Figure 7e: A blue dot is missing in the legend for "ctrl".

Response: we are sorry for the mistake and have added blue dot in the figure.

- Figure S5: Write "without" instead of "w/o".

Response: we have modified it accordingly.

- Figure S10: As this analysis is done on a cohort of Follicular lymphoma, it does not add convincing evidence for a role in KDM3A in leukemia. I therefore suggest to remove it.

Response: we agree with the reviewer's suggestion and have removed it.

- Line 237/238: Refer to Figure S5d instead of S4d

Response: we are sorry for the mistake and have corrected it.